# ANSWER SET CONSISTENCY OF LLMS FOR QUESTION ANSWERING

## ABSTRACT

Large Language Models (LLMs) sometimes contradict themselves when answering factual questions, especially when asked to enumerate all entities that satisfy the question. We formalize such self-contradiction as answer-set inconsistency: Given two enumeration questions whose answers satisfy a set-theoretic relation (equivalence, disjointness, containment, etc.), the LLM generates responses violating the relation. To diagnose this phenomenon, we create a benchmark dataset comprising tuples of enumeration questions over which a variety of set-theoretic relations hold, and propose related metrics to quantify answer-set inconsistency. Our evaluation of several state-of-the-art LLMs reveals pervasive inconsistency across models, even in cases where the LLM can identify the correct relation. This leads us to further analyze potential causes and propose mitigation strategies wherein the LLM is prompted to reason about such relations before answering, which leads to improved answer-set consistency. This work thus provides both a benchmark and a systematic approach for evaluating, explaining, and addressing answer-set inconsistency in LLM question answering, towards deriving practical insights to improve the reliability of LLMs.

## 1 INTRODUCTION

Large Language Models (LLMs) have demonstrated impressive capabilities not only in natural language understanding and generation, but also in question answering and other tasks involving complex reasoning (Tan et al., 2023; Ding et al., 2024; Li et al., 2024; Saxena et al., 2024; Sui et al., 2024). However, in the context of the latter tasks, they are prone to various types of contradiction (Ghosh et al., 2025; Calanzone et al., 2025) since they are not based on computational techniques that guarantee formal notions of consistency, soundness, etc.

One type of inconsistency that LLMs exhibit relates to factual question answering. Specifically, in the context of *enumeration questions*, which ask to list all entities that satisfy a question, the responses across different questions may exhibit inconsistency with respect to evident set-theoretic relations that hold between such questions. Take, for example, the four questions in Table 1. All such questions are enumeration questions: they expect a set of entities as answers. Let $[\![Q]\!]$ denote the expected set of answers for a question $Q$. Among these four questions, we can see that certain set-theoretic relations should be expected to hold, including *equality* ($[\![Q_1]\!] = [\![Q_2]\!] = [\![Q_3]\!] \cup [\![Q_4]\!]$), *containment* ($[\![Q_3]\!] \subseteq [\![Q_1]\!]$, $[\![Q_4]\!] \subseteq [\![Q_1]\!]$, $[\![Q_3]\!] \subseteq [\![Q_2]\!]$, $[\![Q_4]\!] \subseteq [\![Q_2]\!]$, and such relations entailed by equality), *disjointness* ($[\![Q_3]\!] \cap [\![Q_4]\!] = \emptyset$), etc. However, the answers returned by a particular model may not satisfy these relations, even in cases where the model can recognize the correct expected relation. Specifically, let $[\![Q]\!]_M$ denote the set of answers enumerated by model $M$. Then, for example, given $Q_1$ and $Q_3$, when asked what relation holds between their answers, $M$ may correctly recognize the containment of the latter in the former ($[\![Q_3]\!] \subseteq [\![Q_1]\!]$), but still enumerate $[\![Q_1]\!]_M$ and $[\![Q_3]\!]_M$ such that $[\![Q_3]\!]_M \not\subseteq [\![Q_1]\!]_M$, thus contradicting itself.

We formalize this issue as *answer-set consistency*, wherein *the answers for a tuple of factual enumeration questions generated by a particular model do satisfy the set-theoretic relations that are expected to hold for that tuple*. We further consider an *answer-set contradiction* whereby *the model enumerates answers that do not satisfy the set-theoretic relation it itself predicts*. Related topics have been well-studied in database theory literature wherein the notions of *query containment*, *equivalence*, etc., are textbook topics (Abiteboul et al., 1995), with decades of theoretical and practical re-

Table 1: Illustrative example with four enumeration questions forming different relationships (equivalence, containment, disjointness, etc.) with respect to the expected answer sets.

| | |
|---|---|
| $Q_1$ | What are the tributaries of the Madeira River? |
| $Q_2$ | Which rivers and streams flow directly into the Madeira River? |
| $Q_3$ | What are the right-bank tributaries of the Madeira River? |
| $Q_4$ | What are the left-bank tributaries of the Madeira River? |

sults covering a variety of query languages, data models, and reasoning formalisms. Unlike database systems, generative A.I. models are not designed to guarantee satisfaction of such formal relations in the answers they provide. Perhaps for this reason, answer-set (in)consistency has not been well-studied in the context of generative A.I. To the best of our knowledge, the closest work is that of Elazar et al. (2021), which evaluates consistency across paraphrased versions of cloze-style phrases, but these permit only a single answer. Hogan et al. (2025) briefly discuss this issue as "*coherence*", but do not investigate it further. Given that such models are increasingly used to answer users' enumeration questions, we believe the topic merits more analysis.

**Research questions**   In this paper, we address the following research questions (RQs):

**RQ1** To what extent do LLMs produce (in)consistent answer sets for enumeration questions?

**RQ2** Can LLMs recognize the set-theoretic relations that exist between enumeration questions?

**RQ3** Which set-theoretic relations cause the most difficulty for LLMs?

**RQ4** Which key factors cause answer-set inconsistency in LLMs?

**RQ5** Can we mitigate answer-set inconsistency with prompting strategies?

**Contributions.**   This paper describes four main contributions: (1) We highlight and formalize the notion of answer-set consistency for enumeration questions. (2) To quantify answer-set consistency, we develop and release a novel handcrafted benchmark of 600 question quadruples, with 2,400 questions in total, where fixed relations are expected to hold between the questions of each tuple. We further propose measures to quantify answer-set consistency with respect to such a dataset. (3) We present an empirical analysis of 18 state-of-the-art LLMs on the benchmark. (4) We present preliminary prompting strategies to mitigate answer-set inconsistency and evaluate their effectiveness.

Our methodology allows for a systematic analysis of how model size, architecture, and prompting strategy affect the logical coherence of LLM outputs, providing insights into both the strengths and limitations of current models for generating consistent answers to enumeration questions. Our findings reveal that LLMs exhibit significant answer-set inconsistency, the extent of which depends on the particular model and the particular relation (interestingly, newer or bigger models do not universally outperform older or smaller variants). Such inconsistency often occurs even when the model is able to correctly recognize the relation expected to hold. The prompting strategies we propose to mitigate this issue significantly improve consistency across all tested models.

The code, datasets, and results of experiments are available on (anonymous) GitHub (authors, 2025).

## 2   RELATED WORK

**Question answering datasets**   A wide range of datasets have been proposed for different flavors of question answering tasks (e.g., Tan et al. (2023); Wang et al. (2023); Lee & Kim (2024); Singhal et al. (2024); Wang et al. (2024); Zheng et al. (2024); Zhou & Duan (2024); Zhu et al. (2024); Allemang & Sequeda (2025); Ma et al. (2025)). These datasets and evaluation frameworks focus on accuracy with respect to a predefined ground truth. While such benchmarks are essential to verify the correctness and completeness of answers, our focus is rather on the internal consistency of the models, which we see as complementary to these existing datasets.

**Consistency of LLMs**   Various works have addressed different notions of consistency for LLMs.

Ghosh et al. (2025) studies the logical consistency of LLMs in a boolean fact-checking setting. They define the logical consistency of an LLM in terms of the boolean response $LLM(p)$ for some propositional statement $p$ representing a fact. Specifically, they test *negation* ($LLM(\neg p) = \neg LLM(p)$, i.e., the response of the LLM to a negated fact should be the negation of the response to the original fact), *disjunction* ($LLM(p \vee q) = LLM(p) \vee LLM(q)$, i.e., the response to a conjunction of facts is the same as the conjunction of the responses to each fact), and *conjunction* ($LLM(p \wedge q) = LLM(p) \wedge LLM(q)$, as before). The authors further construct complex statements using the combinations of these three boolean operators $\{\neg, \vee, \wedge\}$. Their results showed that consistency improved with an increase in the number of model parameters.

Similarly, Calanzone et al. (2025) investigated whether a fine-tuning approach that integrates neuro-symbolic reasoning can enforce logical consistency in language models. The authors include negation consistency and also whether the LLM can follow implication rules. Empirical results demonstrate that the approach outperforms conventional fine-tuning and external solver-based methods.

While these works address statements in propositional logic, Liu et al. (2024c) investigates logical consistency in LLMs as a property of the relationships between sets of items, rather than isolated predictions. Instead of evaluating responses in isolation, the model is queried across sets of comparisons to assess whether its preferences form a coherent structure. They define consistency through properties like transitivity, order-invariance, and semantic negation, applied across sets of judgments. Their experiments reveal that popular LLMs frequently violate these properties, even on simple domains, and that improving consistency correlates with better downstream task performance.

Jang et al. (2022) propose a behavioral definition of consistency, categorizing it into semantic, logical, and factual types. They introduce BECEL, a benchmark designed to evaluate logical consistency through controlled input changes across several NLP tasks.

Addressing factual questions and statements with a single response, Elazar et al. (2021) evaluates the consistency of (L)LMs across paraphrased versions of cloze-style phrases (i.e., facts with entities masked, such as "___ *is the largest tributary of the Madeira River*"). The authors found notable inconsistency in the models tested at that time. Cohen et al. (2023) propose having one LLM cross-examine another LLM, asking it diverse follow-up questions on the facts it states – for example, by rephrasing the fact and posing it as a question, asking about logical implications of the stated fact, etc. – checking if the response of the latter model remains consistent under cross-examination.

Zhao et al. (2023) investigated whether LLMs can determine semantic equivalence between SQL queries, while Wei et al. (2025) introduced a benchmark to evaluate LLMs' reasoning about program semantics through equivalence checking. These studies, though related, are limited to equivalence relations, without addressing more general or asymmetric semantic relations.

Although the consistency of LLMs with respect to boolean statements, facts, and cloze-style phrases has been studied, the consistency of their responses with respect to questions that permit sets of answers has, to the best of our knowledge, not been explored in depth.

**Query containment**  Given a database $D$ and a structured query $Q$, let $[\![Q]\!]_D$ denote the answers for the query on that database. Given two queries $Q_1$ and $Q_2$, query containment asks if, for any database $D$ (whose schema is compatible with $Q$), it holds that $[\![Q_1]\!]_D \subseteq [\![Q_2]\!]_D$, while *query equivalence* asks whether or not $[\![Q_1]\!]_D = [\![Q_2]\!]_D$ likewise holds for any database (Abiteboul et al., 1995). These problems have been studied for a variety of query languages and database models. While these problems are decidable for simple query formalisms like conjunctive queries under set semantics (akin to queries that only allow relational-style joins), they are undecidable for more expressive query languages (like the relational algebra, full SQL, etc.) (Abiteboul et al., 1995). Referring back to the work of Zhao et al. (2023), LLMs thus cannot decide equivalence for the full SQL language, but it is of interest to know where the limits lie, and how to improve performance.

## 3 ANSWER-SET CONSISTENCY

In this section, we define the notion of answer-set consistency, describe the dataset we create to evaluate it, the metrics we use to quantify it, and the mitigation strategies we propose to address it.

## 3.1 Definition

We say that $Q$ is an *enumeration question* if it has a ground truth answer that is a set, denoted $[\![Q]\!]$ (see Table 1). Let $M$ be a model (e.g., an LLM, potentially primed with a prompt). We denote by $[\![Q]\!]_M$ the answer set generated by model $M$ for question $Q$.

We now define the related notions of *answer-set consistency* and *answer-set contradiction*.

Given two enumeration questions $Q_1$ and $Q_2$ that are equivalent (i.e., $[\![Q_1]\!] = [\![Q_2]\!]$), we say that a model $M$ is *answer-set consistent* with respect to the equivalence of $(Q_1, Q_2)$ if and only if $[\![Q_1]\!]_M = [\![Q_2]\!]_M$. If this property does not hold, we call the model *answer-set inconsistent*. We likewise define answer-set (in)consistency for other relations, including *containment* (given $[\![Q_1]\!] \subseteq [\![Q_2]\!]$, check $[\![Q_1]\!]_M \subseteq [\![Q_2]\!]_M$), *disjointness* (given $[\![Q_1]\!] \cap [\![Q_2]\!] = \emptyset$, check $[\![Q_1]\!]_M \cap [\![Q_2]\!]_M = \emptyset$) and *overlap* (given $[\![Q_1]\!] \cap [\![Q_2]\!] \neq \emptyset$, check $[\![Q_1]\!]_M \cap [\![Q_2]\!]_M \neq \emptyset$).

We do not need ground-truth answer sets for questions in order to analyze answer-set consistency.

**Example 3.1.** Considering the four questions in Table 1, even without the ground-truth answer sets, we see that $[\![Q_1]\!] = [\![Q_2]\!]$ (equivalence), since *tributaries* of a river are defined as *rivers and streams* that *flow* directly into that river. Similarly, we see that $[\![Q_3]\!] \subseteq [\![Q_1]\!]$ and $[\![Q_4]\!] \subseteq [\![Q_1]\!]$ (containment) since both left- and right-bank tributaries should be included in the answer for all tributaries. Furthermore, we see that $Q_3 \cap Q_4 = \emptyset$ (disjointness), as right-bank and left-bank tributaries are mutually exclusive. Other implied relations exist, for example, $[\![Q_3]\!] \subseteq [\![Q_2]\!]$ (containment), $[\![Q_3]\!] \cap [\![Q_2]\!] \neq \emptyset$ (overlap). Hence, for example, if a model would give overlapping answers for $Q_3$ and $Q_4$, we would deem it answer-set inconsistent with respect to the disjointness of $(Q_3, Q_4)$.

The presented definition is agnostic to the open world assumption, or closed world assumption, since we are solely interested in the relationship between two answers, regardless of whether elements not included in the answer are treated as false, or unknown to the model. However, under the open world assumption, a model may return empty answers, which are trivially consistent; for this reason, in our experiments, we will handle cases in which the models returns empty answers separately.

Next we define *answer-set contradictions*. Given two enumeration questions $Q_1$ and $Q_2$, let $[\![Q_1, Q_2]\!]_M \subseteq \{\text{E}, \text{C}, \text{D}, \text{O}\}$ denote the prediction of model $M$ for which binary relations hold for $(Q_1, Q_2)$: Equivalence, Containment, Disjointness, or Overlap. We represent the prediction as a set since multiple relations can hold. We say that a model $M$ gives an *answer-set contradiction* if it predicts a relation $R \in [\![Q_1, Q_2]\!]_M$ that does not hold for $[\![Q_1]\!]_M$ and $[\![Q_2]\!]_M$. In other words, it predicts a relation for $Q_1$ and $Q_2$ that its own answers for $Q_1$ and $Q_2$ do not respect. This is a self-contradiction: the model contradicts itself, rather than the expectation relative to a ground truth.

**Example 3.2.** Considering the first two questions in Table 1, if a model predicts that these two questions are equivalent (E), but gives answer sets $[\![Q_1]\!]_M$ and $[\![Q_2]\!]_M$ such that $[\![Q_1]\!]_M \neq [\![Q_2]\!]_M$, this represents an answer-set contradiction with respect to equivalence.

For reasons of space, we delegate discussion of *answer-set contradictions* to Appendix F.

## 3.2 Dataset

In order to evaluate the *answer-set consistency* of various LLMs, and to address our research questions, we require a collection of pairs of questions that satisfy relations such as equivalence, containment, etc. Given that we know of no such existing dataset, we design and construct one.

We aim for questions whose ground-truth answers are objective and non-empty. We further aim for questions with between 2 and 100 results: having 2 results enables us to test (strict, non-empty) containment over such questions, and we set a limit of 100 results to avoid context-window issues when testing smaller models. Example questions meeting these criteria are *Which countries are members of the European Union?*, *What are the tributaries of the Madeira River?*, etc. Such questions can be found in knowledge graph question answering (KGQA) datasets, which provide a good starting point, since they contain mostly objective enumeration questions, and we can use the queries provided to filter cases with too few or too many results. Some datasets further provide paraphrases that cover equivalence. Unfortunately, only one KGQA dataset, QAWIKI (Moya Loustaunau & Hogan, 2025), provides containment relations, and only a few.

We constructed our dataset, which we call the Answer-Set Consistency Benchmark (ASCB), from three base KGQA datasets: LC-QUAD 2.0 (Dubey et al., 2019), QALD (Usbeck et al., 2024), and QAWIKI (Moya Loustaunau & Hogan, 2025). From each dataset, we evaluated the structured queries associated with questions to ensure that they are enumeration questions and to test that they satisfy the cardinality bounds. A preliminary filtering step was conducted on the questions contained in the LC-QUAD 2.0 and QALD datasets using LLMs. This step aimed to remove questions that were not aligned with our criteria and to generate new questions that conform to the logical relations under evaluation. The complete procedure for filtering and question generation is described in Appendix B.2. This created a candidate list of base questions, which we manually reviewed and filtered down to a smaller set of selected questions that best meet our criteria, as well as additional desiderata prioritizing the "crispness"[1], diversity, fluency, and objectiveness of questions. We enriched this candidate set with available paraphrases to cover the case of equivalence, creating a set of pairs of equivalent questions of the form $(Q_1, Q_2)$ such that $[\![Q_1]\!] = [\![Q_2]\!]$.

We next looked to add examples for (strict) containment to our dataset. To reduce manual effort, rather than start from scratch, we used LLMs to extend the equivalent question pairs $(Q_1, Q_2)$ created previously to suggest a third query $Q_3$ such that $[\![Q_3]\!] \subseteq [\![Q_1]\!]$ (and, by implication, $[\![Q_3]\!] \subseteq [\![Q_2]\!]$). During this process, it required relatively little additional effort to define a fourth query, $Q_4$, that captures the answers of $Q_1$ that $Q_3$ does not, such that $[\![Q_4]\!] = [\![Q_1]\!] \setminus [\![Q_3]\!]$, providing an additional example not only for strict containment, but also disjointness.

Thus our initial dataset consisted of quadruples of questions $(Q_1, Q_2, Q_3, Q_4)$, where Table 1 exemplifies one of the quadruples in our dataset. From such a quadruple of questions, there are 12 primary relations, where, for clarity, we call the superset containment *broader*, and its inverse subset containment *narrower*. These relations are *equivalence* for $(Q_1, Q_2)$ and $(Q_2, Q_1)$; *broader* for $(Q_1, Q_3)$, $(Q_1, Q_4)$, $(Q_2, Q_3)$ and $(Q_2, Q_4)$; *narrower* for $(Q_3, Q_1)$, $(Q_3, Q_2)$, $(Q_4, Q_1)$ and $(Q_4, Q_2)$, and *disjointness* for $(Q_3, Q_4)$ and $(Q_4, Q_3)$. There are also further implicit relations, such as broader and narrower implied by equivalence, and overlaps implied by broader, narrower and equivalence, assuming non-empty answer sets; for example, given the broader relation for $(Q_1, Q_3)$ (whereby $[\![Q_1]\!] \supseteq [\![Q_3]\!]$), and assuming non-empty answer sets ($[\![Q_1]\!] \neq \emptyset$, $[\![Q_3]\!] \neq \emptyset$), this implies that overlap holds for the same question pair ($[\![Q_1]\!] \cap [\![Q_3]\!] \neq \emptyset$).

These candidate questions were manually revised, pruned, modified, etc., to ensure a high-quality dataset, resulting in 150 question quadruples from each dataset. We further added an additional source, which we call SYNTHETIC, of questions generated from scratch by LLMs, the generation process is detailed in Appendix B. It is important to note that in many cases, the questions extracted merely served as "inspiration" for the final quadruple, even for the base query: the questions in many cases were heavily modified. Furthermore, suggestions by LLMs for $Q_3$ and $Q_4$, though useful, often did not satisfy the formal relations expected[2], and needed to be modified. As a final step, we used LLMs to revise the quadruples and suggest improvements for phrasing, corrections, etc., which were revised manually and applied if deemed suitable. The manual revision, curation, and modification of questions were conducted by three of the authors.

The resulting ASCB dataset (available online (authors, 2025)) comprises 600 such quadruples of handcrafted questions in English, with 2,400 questions in total.

### 3.3 EVALUATION TASKS AND MITIGATION STRATEGIES

To evaluate the answer-set consistency of LLMs using our ASCB, we perform three distinct tasks, the latter two of which involve mitigation strategies to try to achieve better answer-set consistency (rather than the completeness or correctness of answer sets). For all such tasks, the LLMs under test are configured to use the lowest possible temperature. The prompts used are given in Appendix A.

**Task 1: Base evaluation.** Our first task evaluates the base answer-set consistency of the LLM under test, and involves two subtasks.

---

[1]This relates to the results forming a crisp set: some questions, such as "*What are the jobs in finance?*" in LC-QuAD 2.0, do not clearly define a "crisp" base set for results, and are excluded.

[2]Often the suggestions for $Q_3$ and $Q_4$ did not form a true dichotomy, for example: *What counties of North Dakota use the Mountain Time Zone?* and *What counties of North Dakota use a timezone other than the Mountain Time Zone?* is not a true dichotomy as some counties are in multiple time zones.

Table 2: The relations considered for experiments: their notation, definition and meaning

| Notation | Definition | Meaning |
|----------|-----------|---------|
| $E_{1,2}$ | $[\![Q_1]\!] = [\![Q_2]\!]$ | *Equivalence*: Results of $Q_1$ are the same as those of $Q_2$ |
| $N_{3,1}$ | $[\![Q_3]\!] \subseteq [\![Q_1]\!]$ | *Narrower*: Results of $Q_3$ are contained in those of $Q_1$ |
| $N_{4,1}$ | $[\![Q_4]\!] \subseteq [\![Q_1]\!]$ | *Narrower*: Results of $Q_4$ are contained in those of $Q_1$ |
| $D_{3,4}$ | $[\![Q_3]\!] \cap [\![Q_4]\!] = \emptyset$ | *Disjointness*: No result of $Q_3$ is a result of $Q_4$ |
| $E_{4,1\setminus 3}$ | $[\![Q_4]\!] = [\![Q_1]\!] \setminus [\![Q_3]\!]$ | *Equivalence*: Results obtained by removing the answer set of $Q_3$ from that of $Q_1$ are the same as those of $Q_4$ |
| $E_{1,*}$ | $[\![Q_1]\!] = [\![Q_1^*]\!]$ | *Equivalence*: Results of $Q_1$ are the same as those of $Q_1$ posed in a different context at a a different time |

In Task 1.1, *Classification*, the models are presented pairs of questions $(Q_i, Q_j)$ taken from each quadruple and asked to identify the *first* set-theoretic relation that holds, in the following order, from the answer set of $Q_i$ to that of $Q_j$: equivalence ($[\![Q_i]\!] = [\![Q_j]\!]$), contained by ($[\![Q_i]\!] \subseteq [\![Q_j]\!]$), contains ($[\![Q_i]\!] \supseteq [\![Q_j]\!]$), disjointness ($[\![Q_i]\!] \cap [\![Q_j]\!] = \emptyset$) and overlap ($[\![Q_i]\!] \cap [\![Q_j]\!] \neq \emptyset$). Although any pair of questions must satisfy at least one such relation, we further add an *unknown* option ('idk') in case the model cannot confidently determine the relation.

In Task 1.2, *Enumeration*, all 2,400 questions are posed to the models independently, each within its own isolated context, requesting the LLM to enumerate all answers for the question. The prompt further instructs the model to return an exhaustive list separated by '|', to avoid additional text, to return 'idk' in case it cannot answer, to return 'no answer' if there is no answer, and to expand acronyms and use full names whenever possible. The corresponding answers are then collected.

**Task 2: Classification-then-Enumeration (CtE).** Following initial experiments on Task 1, we investigated a mitigation strategy that operates within a conversational context. The model is first asked to identify the relationship between the questions, and is then prompted to enumerate their answers. Our hypothesis is that this will improve answer-set consistency by allowing the model to first reason about the relation that the answer sets should satisfy before enumerating them.

**Task 3: Oracle.** We first run Task 1.1, and if we detect an answer-set inconsistency with respect to a given relation for a question pair $(Q_i, Q_j)$, we tell the LLM the correct relation that holds, and request it to enumerate answers again. This task is an ideal version of Task 2, as it assumes an oracle that knows what relation holds between the questions. This will give us insights into what the model could achieve in Task 2 if it always classifies the relation correctly.

**Question and relations considered.** Our quadruples provide a large number of potential binary relations to test. However, many such relations are redundant. Thus, we only consider each pair of questions once (skipping symmetric and inverse relations), and only test for the primary relation. The selected relations are presented in Table 2. The relations $E_{1,2}$, $N_{3,1}$, $N_{4,1}$, and $D_{3,4}$ cover equivalence, containment (narrower) and disjointness. We also test an $n$-ary relation $E_{4,1\setminus 3}$ for an answer set constructed from set difference as an example of a more complex task. Finally, to establish a referential result for non-determinism of the model over time, we test $E_{1,*}$, which runs the same question $Q_1$ at different times, in different contexts, which we will use as a control later.

## 3.4 Evaluation Measures

We present various measures to evaluate the answer-set consistency of LLMs with respect to the previous tasks. The first two address the performance for enumeration, while the second addresses the performance for classification.

**Classification accuracy.** We first quantify the ability of LLMs to correctly classify the set-theoretic relation that holds between the answer-sets of the questions. For each relation $R \in \{E_{1,2}, N_{3,1}, N_{4,1}, D_{3,4}, E_{4,1\setminus 3}\}$, the *classification accuracy* of a model $M$ over $n$ test instances ($n = 600$ for ASCB) is defined as $R^{\text{ACC}}(M) = \frac{\text{\# correct classifications of } R \text{ by } M}{n}$. We further denote by $\text{ACC}(M)$ the average of the accuracy of the five aforementioned relations for the model $M$.

**Consistency rates.** For each pair of enumerated answer sets, we evaluate whether or not the six expected relations $R \in \{E_{1,2}, N_{3,1}, N_{4,1}, D_{3,4}, E_{4,1\backslash 3}, E_{1,*}\}$ are satisfied. Each such binary relation $R$ is checked independently over $n$ test instances, where we say that an instance is consistent for $R$ if the enumerated answers for the questions satisfy $R$. The *consistency rate* of $M$ for $R$ is then defined as $R^{\text{CON}}(M) = \frac{\text{\# consistent answer sets for } R \text{ by } M}{n}$. Here we exclude empty answer sets and responses of "idk", which are reported separately. We further denote by $\text{CON}(M)$ the average consistency rate of $M$ over the five relations excluding $E_{1,*}$ (which is intended as a control).

**Jaccard similarity.** The consistency rates consider each $(Q_i, Q'_i, \star_i)$ as a discrete result (1 for correct, or 0 for incorrect) – irrespective of how close the answer sets of the two questions are to satisfying the relation. To complement the first measure, we consider a second continuous measure that captures the degree to which an instance is satisfied. Specifically, we use the well-known *Jaccard similarity*, defined for sets $S_1$ and $S_2$ as $J(S_1, S_2) = \frac{|S_1 \cap S_2|}{|S_1 \cup S_2|}$. For a given model $M$, and test instance $(Q_i, Q_j)$, we define the similarity of the test instance as $J(\llbracket Q_i \rrbracket_M, \llbracket Q_j \rrbracket_M)$. The Jaccard similarity of a model $M$ for a relation $R \in \{E_{1,2}, D_{3,4}, E_{4,1\backslash 3}, E_{1,*}\}$ over $n$ test instances is then defined as $R^{\text{SIM}}(M) = \frac{\text{sum of similarity of all test instances by } M}{n}$, i.e., as the average similarity over $R$. Empty answer sets and "idk" are excluded from these results and are reported separately. We apply this measure for relations involving equivalence ($E_{1,2}$, $E_{4,1\backslash 3}$, $E_{1,*}$) and disjointness ($D_{3,4}$), where a score close to 1 is good in the former case, and poor in the latter case.

**Empty rates.** We also measure the percentage of the 2,400 questions that return "idk" or an empty answer, denoted as $\%\text{IDK}$.

**Hypotheses and significance testing.** We propose two hypotheses: ($H_1$) The **CtE** and **Oracle** strategies yield less answer-set inconsistency than **Base**. ($H_2$) LLMs with better general performance produce more consistent responses.

To test statistical significance, we apply the one-sided McNemar test (Lachenbruch, 2014), which is suitable for paired nominal data. We adopt a significance level of $\alpha = 0.05$. The null hypothesis of no improvement is rejected if the one-sided p-value satisfies $p < 0.05$. In such cases, we conclude that the alternative strategy or model yields a statistically significant improvement in consistency.

**Stochasticity control.** The causes of answer-set inconsistency in LLMs can be attributed to two main factors. (1) *Stochasticity* in the generation process, such as during token sampling, or latent variability, leading the model to produce different outputs for the same query across runs. (2) *Semantic misunderstanding*, where the model misinterprets or overlooks the logical or semantic relations among queries, resulting in violations of restrictions. We introduce $E_{1,*}$ precisely to estimate and control for the effects of (1), which varies across models. Further discussion of the causes behind LLM inconsistencies can be found in Appendix G.

The impact specifically of the two factors on the answer-set inconsistency of model $M$ for $E_{1,2}$ can be assessed by comparing $E_{1,2}^{\text{CON}}(M)$ with $E_{1,*}^{\text{CON}}(M)$ and $E_{1,2}^{\text{SIM}}(M)$ with $E_{1,*}^{\text{SIM}}(M)$: a large gap indicates (2) plays a smaller role, and a small gap indicates (2) plays a larger role. On the other hand, we can compare the difference of consistency for equivalence relations like $E_{1,*}$, $E_{1,2}$ with other relations like $N_{3,1}$, $N_{4,1}$, $D_{3,4}$ where (assuming stocasticity plays a similar role for such relations) a large gap indicates (2) plays a larger role, and a small gap indicates (2) plays a smaller role.

## 4 RESULTS

We now present the results of our experiments on 18 LLMs from the DeepSeek, Gemini, Grok, GPT, Llama, and Mistral families. For all models, the temperature was set to the lowest possible value (zero, if possible). LLMs are ranked in ascending order based on their Global Average scores reported by White et al. (2025) from $A$ (worst) to $R$ (best); open-source models not included in this ranking are loosely positioned alongside models of similar parameter size. The results are reported for the overall dataset, that is, the dataset obtained by merging the four sources presented in Section 3.2. Results for individual datasets are available (anonymously) on GitHub (authors, 2025). For reasons of space, we provide figures that help to identify trends in these results in Appendix C.

## 4.1 CLASSIFICATION TASK

Appendix D reports the accuracy of different LLMs on the relation classification task. The highest accuracy score in each column is indicated in bold. The results reveal substantial variability among the evaluated models. Smaller-scale models such as Llama-3.1-8b exhibit poor performance across all relations, with accuracy often below 20%. Similarly, GPT-oss-20b, GPT-4.1-nano, and Mistral-small:24b perform inconsistently, though notably the relations they struggle with sometimes differ. In contrast, larger models such as Gemini-2.5-pro, GPT-5-nano, GPT-5, GPT-o3, and Grok-3-mini achieve accuracies consistently above (or close to) 90% across all relations. Among the evaluated relations, $E_{1,2}$ and $D_{3,4}$ are the least challenging overall, while $N_{4,1}$ emerges as the most challenging, revealing the limitations of current models in reliably capturing the containment relation.[3] GPT-5 demonstrates the best performance over all relations, followed closely by Gemini-2.5-pro.

## 4.2 ANSWER-SET CONSISTENCY

Table 3 lists the results for answer-set consistency, considering the control relation, all five test relations, and the measures of consistency rate and Jaccard similarity. Recall that, exceptionally for $D_{3,4}^{\mathrm{SIM}}$, a score lower than 0 for Jaccard similarity is better as the answer sets should be disjoint.

We see high rates of answer-set inconsistency for all relations, including even the control relation. The small gap between $E_{1,*}$ and $E_{1,2}$ suggests that stochasticity plays an important role in answer-set inconsistency, even though temperature was lowered as much as possible in all such cases, this phenomenon aligns with prior research demonstrating that LLMs produce inconsistent outputs even with identical inputs. The literature identifies four primary sources of this variability: decoding randomness Ackley et al. (1985); Li et al. (2025); Atil et al. (2024); Renze (2024), computational nondeterminism Yuan et al.; Masoudnia & Ebrahimpour (2014); Dao et al. (2022); Atil et al. (2024), order sensitivity Vaswani et al. (2017); Liu et al. (2024a); Lu et al. (2022), and data-level conflicts Xu et al. (2024); Nakshatri et al. (2025); Xie et al. (2023). Beyond these inherent stochastic factors, we observe additional systematic errors including terminological inconsistency and knowledge gaps that lead to incomplete outputs (see Appendix G and H for more discussion). However, the gap between $E_{1,2}$, $E_{1,*}$ and containment / ternary relations suggests that semantic misunderstanding is also a key cause of inconsistency for these latter relations in particular.

The most inconsistent relation is the ternary relation of $E_{4,1\backslash 3}$, with the most consistent relation overall being the disjointness relation $D_{3,4}$. The best model in terms of average consistency across all relations for the base case is GPT-5 ($\sim$57%), with the important caveat that it exhibits a high %IDK rate ($\sim$32%). Models such as Grok-3-mini and Gemini-2.5-flash arguably perform better, with lower average consistency of $\sim$48% and $\sim$46%, resp., but also lower %IDK values of $\sim$8% and $\sim$5%, respectively. There are notable improvements for the mitigation strategy Classify-then-Enumerate (CtE), which (surprisingly) even outperforms the Oracle in many cases, due to good classification accuracy (see Appendix D), and perhaps due to forcing the LLM itself to reason about the questions when classifying their relation. We also observed that the %IDK values for CtE are generally higher than Base and Oracle. This suggests that under this strategy, LLMs tend to adopt a safer approach by answering "idk" when uncertain, which may explain why CtE outperforms the other two strategies. Improvement with CtE is not universal, however: CtE sometimes performs worse than Base due to the model being unable to classify the relation (as is the case for $E_{4,1\backslash 3}$ in GPT-4.1-nano, which answers idk for each case when asked to identify the relation), or due to the model misinterpreting the more complex prompts (e.g, GPT-5-nano often continues to return a relation for $E_{4,1\backslash 3}$ *after* classification, when later asked to enumerate results).

## 4.3 HYPOTHESIS TESTING

In Appendix E, we present an analysis of the statistical significance of our results. Regarding hypothesis $H_1$, that "The **CtE** and **Oracle** strategies yield less answer-set inconsistency than **Base**", this is confirmed by a $p$-value $< 0.001$ for almost all models for both strategies. Regarding hypothesis $H_2$, the mitigation strategies significantly affect consistency and relative model rankings.

---

[3]Regarding why $N_{4,1}$ is more challenging than $N_{3,1}$, $N_{3,1}$ is based on $Q_3$, and $N_{4,1}$ is based on $Q_4$, where $Q_4$ questions tend to negate the restriction that $Q_3$ adds over $Q_1$, and this negation appears more challenging.

Table 3: Per-relation consistency (0-100 scale) for each model and strategy (Str).

| ID | Model | Str | $E_{1,*}^{\mathrm{CON}}$ | $E_{1,*}^{\mathrm{SIM}}$ | $E_{1,2}^{\mathrm{CON}}$ | $N_{3,1}^{\mathrm{CON}}$ | $N_{4,1}^{\mathrm{CON}}$ | $D_{3,4}^{\mathrm{CON}}$ | $E_{4,1\backslash3}^{\mathrm{CON}}$ | CON | $E_{1,2}^{\mathrm{SIM}}$ | $D_{3,4}^{\mathrm{SIM}}$ | $E_{4,1\backslash3}^{\mathrm{SIM}}$ | %IDK |
|----|-------|-----|------|------|------|------|------|------|------|------|------|------|------|------|
| A | Llama-3.1-8b | Base | 2.92 | 21.52 | 1.67 | 3.00 | 1.17 | 84.33 | 0.00 | 18.03 | 16.67 | 3.29 | 6.44 | 1.66 |
|   |  | CtE | –"– | –"– | 16.33 | **100.00** | 33.83 | 68.17 | 5.00 | 44.67 | 56.03 | 9.74 | 38.99 | 25.08 |
|   |  | Ora. | –"– | –"– | 12.67 | 15.50 | 12.33 | 61.83 | 1.83 | 20.83 | 45.49 | 11.26 | 24.14 | 0.21 |
| B | GPT-oss-20b | Base | 22.71 | 44.44 | 21.67 | 38.00 | 32.50 | 83.00 | 10.00 | 37.03 | 44.13 | 7.75 | 29.27 | 12.54 |
|   |  | CtE | –"– | –"– | 67.17 | **100.60** | 73.17 | 84.67 | 38.00 | 72.60 | 79.39 | 10.73 | 60.99 | 37.04 |
|   |  | Ora. | –"– | –"– | 58.33 | 80.67 | 92.33 | 95.33 | 62.00 | 77.73 | 76.50 | 3.97 | 78.64 | 13.33 |
| C | gpt-4.1-nano | Base | 58.52 | 68.23 | 27.83 | 45.67 | 37.17 | 62.00 | 4.00 | 35.33 | 51.29 | 19.15 | 20.37 | 12.58 |
|   |  | CtE | –"– | –"– | **97.33** | 71.67 | 56.67 | 38.17 | 0.00 | 52.77 | **97.57** | 61.49 | 0.17 | **80.87** |
|   |  | Ora. | –"– | –"– | 54.67 | 59.17 | 60.17 | 61.83 | 21.50 | 51.47 | 62.37 | 22.57 | 35.09 | 19.37 |
| D | Mistral-small:24b | Base | 43.01 | 64.10 | 44.07 | 50.34 | 45.25 | 50.85 | 1.53 | 38.41 | 63.99 | 34.47 | 20.57 | 29.19 |
|   |  | CtE | –"– | –"– | 84.50 | **100.00** | 80.67 | 79.83 | 42.33 | 77.47 | 92.57 | 17.74 | 63.71 | 33.58 |
|   |  | Ora. | –"– | –"– | 72.00 | 74.67 | 90.00 | 95.83 | 50.17 | 76.53 | 82.96 | 1.48 | 67.82 | 15.54 |
| E | Llama-3.1-70b | Base | 20.33 | 48.82 | 20.67 | 28.83 | 23.83 | 71.00 | 2.83 | 29.43 | 47.12 | 10.06 | 20.67 | 3.33 |
|   |  | CtE | –"– | –"– | 80.33 | **100.00** | 86.33 | 93.50 | 56.17 | 83.27 | 90.71 | 2.95 | 81.20 | 26.91 |
|   |  | Ora. | –"– | –"– | 61.00 | 72.67 | 82.17 | 93.50 | 50.67 | 72.00 | 75.29 | 3.37 | 76.35 | 5.04 |
| F | Gemini-2.0-flash | Base | 48.03 | 70.08 | 32.67 | 41.50 | 35.33 | 62.67 | 5.00 | 35.43 | 60.08 | 13.00 | 29.14 | 0.42 |
|   |  | CtE | –"– | –"– | 88.17 | **100.00** | 94.50 | 64.33 | 53.83 | 80.17 | 93.84 | 27.82 | 62.68 | 40.58 |
|   |  | Ora. | –"– | –"– | 79.00 | 84.17 | 92.00 | 96.67 | 73.83 | 85.13 | 82.33 | 1.37 | 84.15 | 2.96 |
| G | gpt-4.1-mini | Base | 28.02 | 53.84 | 33.67 | 48.17 | 38.33 | 56.33 | 5.50 | 36.40 | 62.96 | 15.74 | 28.12 | 4.83 |
|   |  | CtE | –"– | –"– | 89.83 | 64.00 | 94.83 | 77.17 | 64.00 | 77.97 | 95.73 | 21.83 | 72.41 | 23.25 |
|   |  | Ora. | –"– | –"– | 77.50 | 87.67 | 96.33 | 89.67 | 68.83 | 84.00 | 88.81 | 6.10 | 79.53 | 8.75 |
| H | GPT-4o | Base | 48.98 | 52.09 | 45.17 | 53.33 | 47.00 | 62.83 | 6.17 | 42.90 | 65.34 | 26.69 | 26.12 | 29.79 |
|   |  | CtE | –"– | –"– | **97.33** | 99.33 | 98.33 | 36.33 | 30.67 | 72.40 | 58.67 | 62.79 | 35.08 | 66.66 |
|   |  | Ora. | –"– | –"– | 85.33 | 91.83 | 94.83 | 86.50 | 67.83 | 85.26 | 73.43 | 12.09 | 58.79 | 33.88 |
| I | gpt-4.1 | Base | 28.73 | 59.49 | 39.67 | 49.83 | 38.67 | 63.33 | 10.17 | 40.33 | 67.71 | 11.18 | 35.81 | 3.79 |
|   |  | CtE | –"– | –"– | 91.83 | 70.33 | 97.50 | 87.00 | 74.83 | 84.30 | 97.54 | 11.74 | **84.84** | 13.75 |
|   |  | Ora. | –"– | –"– | 82.50 | 91.83 | 94.50 | 95.33 | 74.83 | 87.80 | 92.07 | 3.09 | 83.64 | 6.62 |
| J | Grok-3-mini | Base | 37.67 | 63.09 | 34.33 | 52.33 | 43.83 | 87.67 | 23.00 | 48.23 | 63.46 | 5.61 | 44.78 | 8.12 |
|   |  | CtE | –"– | –"– | 90.83 | **100.00** | 90.00 | 86.67 | 66.83 | 86.87 | 96.39 | 12.77 | 78.69 | 35.08 |
|   |  | Ora. | –"– | –"– | 88.17 | 92.67 | **98.50** | 95.50 | 81.00 | 91.17 | 93.63 | 4.00 | 79.16 | 13.46 |
| K | DeepSeek-V3.1 | Base | 38.81 | 60.82 | 32.50 | 45.50 | 40.17 | 56.00 | 7.33 | 36.30 | 57.03 | 16.61 | 25.73 | 11.38 |
|   |  | CtE | –"– | –"– | 94.50 | 84.67 | 91.17 | 67.33 | 54.17 | 78.37 | 97.22 | 31.01 | 62.74 | 30.67 |
|   |  | Ora. | –"– | –"– | 81.67 | 88.83 | 95.00 | 94.67 | 71.00 | 86.23 | 91.17 | 3.96 | 79.80 | 10.00 |
| L | Gemini-2.5-flash | Base | 37.69 | 64.85 | 33.50 | 49.50 | 41.50 | 84.50 | 21.83 | 46.17 | 61.01 | 3.60 | 43.82 | 5.33 |
|   |  | CtE | –"– | –"– | 89.83 | **100.00** | 96.83 | 90.67 | 84.83 | **92.43** | 93.83 | 8.82 | 83.92 | 31.96 |
|   |  | Ora. | –"– | –"– | 89.17 | 90.17 | 95.33 | 95.67 | 85.50 | 91.17 | 92.56 | 4.08 | 83.49 | 8.00 |
| M | GPT-5-nano | Base | 33.48 | 42.09 | 59.00 | 64.00 | 63.17 | 67.67 | 17.33 | 54.23 | 72.28 | 29.88 | 38.52 | 41.79 |
|   |  | CtE | –"– | –"– | 77.83 | **100.00** | 76.83 | 19.50 | 0.50 | 54.93 | 78.16 | **80.39** | 7.82 | 31.00 |
|   |  | Ora. | –"– | –"– | 84.17 | 83.17 | 97.17 | 77.83 | 53.33 | 79.13 | 89.65 | 21.40 | 58.45 | 41.25 |
| N | DeepSeek-reasoner | Base | 29.95 | 51.86 | 18.67 | 40.17 | 30.17 | 82.00 | 10.83 | 36.37 | 46.10 | 3.96 | 31.15 | 4.21 |
|   |  | CtE | –"– | –"– | 78.50 | 75.33 | 85.33 | 69.67 | 51.50 | 72.07 | 82.40 | 29.66 | 56.37 | 28.54 |
|   |  | Ora. | –"– | –"– | 68.33 | 75.67 | 86.33 | 93.00 | 58.33 | 76.33 | 82.21 | 5.37 | 79.55 | 6.25 |
| O | Gemini-2.5-pro | Base | 36.88 | 66.44 | 31.00 | 43.67 | 38.83 | 81.50 | 19.50 | 42.90 | 63.14 | 3.25 | 46.98 | 3.33 |
|   |  | CtE | –"– | –"– | 85.33 | **100.00** | 88.00 | 93.00 | 71.33 | 87.53 | 89.78 | 6.22 | 79.02 | 33.00 |
|   |  | Ora. | –"– | –"– | 77.33 | 80.17 | 92.17 | **97.83** | 74.83 | 84.47 | 81.97 | 2.06 | 80.89 | 10.50 |
| P | GPT-5-mini | Base | 63.15 | 76.22 | 65.33 | 63.83 | 68.17 | 61.50 | 14.33 | 54.63 | 77.18 | 35.93 | 37.25 | 47.17 |
|   |  | CtE | –"– | –"– | 88.00 | **100.00** | 75.00 | 67.33 | 34.00 | 72.87 | 92.58 | 32.67 | 44.64 | 55.08 |
|   |  | Ora. | –"– | –"– | 86.83 | 91.67 | 95.00 | 73.67 | 54.50 | 80.33 | 91.04 | 26.17 | 55.03 | 50.04 |
| Q | GPT-o3 | Base | 31.42 | 58.92 | 34.41 | 50.68 | 38.64 | 86.61 | 19.15 | 45.90 | 62.46 | 5.53 | 43.83 | 8.98 |
|   |  | CtE | –"– | –"– | 82.00 | 88.33 | 91.33 | 92.50 | 74.67 | 85.77 | 93.24 | 6.74 | 83.97 | 26.58 |
|   |  | Ora. | –"– | –"– | 73.67 | 92.00 | 94.50 | 97.33 | 77.17 | 86.93 | 82.18 | 2.44 | 80.82 | 13.88 |
| R | GPT-5 | Base | 58.59 | 76.89 | 61.00 | 64.17 | 65.00 | 73.50 | 21.67 | 57.07 | 78.16 | 22.11 | 48.16 | 32.00 |
|   |  | CtE | –"– | –"– | 92.33 | **100.00** | 96.50 | 70.67 | 59.83 | 83.87 | 96.63 | 29.20 | 66.83 | 47.50 |
|   |  | Ora. | –"– | –"– | 90.67 | 94.00 | 98.17 | 79.00 | 65.33 | 85.43 | 96.11 | 21.00 | 69.17 | 34.59 |

Table 4 shows significant positive correlations between $D_{3,4}^{\mathrm{CON}}$, $E_{4,1\backslash3}^{\mathrm{CON}}$, and $D_{3,4}^{\mathrm{SIM}}$ and the average score of the models on the external benchmark (White et al., 2025). Models not in this benchmark are excluded. In particular, $E_{4,1\backslash3}$ exhibits strong positive correlations across both measures. Since $E_{4,1\backslash3}$ is the most challenging relation to identify, this finding supports $H_2$, suggesting that consistency is more pronounced in the more complex answer-set tasks.

Table 4: Pearson correlations between external model scores and reasoning performance metrics. Asterisks indicate statistical significance: * $p < 0.05$, ** $p < 0.001$.

|  | $E_{1,2}^{\text{CON}}$ | $N_{3,1}^{\text{CON}}$ | $N_{4,1}^{\text{CON}}$ | $D_{3,4}^{\text{CON}}$ | $E_{4,1\backslash 3}^{\text{CON}}$ | $\text{CON}()$ | $E_{1,2}^{\text{SIM}}$ | $D_{3,4}^{\text{SIM}}$ | $E_{4,1\backslash 3}^{\text{SIM}}$ |
|---|---|---|---|---|---|---|---|---|---|
| $r$ | 0.328 | 0.410 | 0.406 | 0.650* | 0.821** | 0.682* | 0.425 | -0.220 | 0.840** |
| $p$ | 0.252 | 0.145 | 0.149 | 0.012 | <0.001 | 0.007 | 0.130 | 0.449 | <0.001 |

## 5 DISCUSSION

We have highlighted the phenomenon of answer-set inconsistency in LLMs, proposed a dataset to evaluate it, defined various measures to quantify it, and presented the results for 18 LLMs.

**Research questions**   We address the RQs presented in the introduction:

**RQ1** LLMs exhibit high degrees of answer-set inconsistency for enumeration questions, with the particular degree depending on the model and relation (see Table 3).

**RQ2** Contemporary large LLMs can recognize the set-theoretic relations that hold between enumeration questions with accuracy often about 90%, though smaller/older models struggle with many types of relations (see Appendix D).

**RQ3** Binary equivalence relations appear to be the easiest relations for the LLMs to reason about, where they struggle with containment and $n$-ary relations (see Appendix D and Table 3).

**RQ4** Based on our control ($E_{1,*}$), much of the answer-set inconsistency for equivalence relations is due to the stochastic nature of LLMs, whereas semantic misunderstanding plays a more dominant role for containment, disjointness and, $n$-ary relations (Table 3).

**RQ5** Answer-set inconsistency can be mitigated (i.e., improved by a wide margin, with statistical significance) by prompting strategies that ask the LLM to reason about the set-theoretic relations that hold between enumeration questions and their answer sets (Table 3).

The performance of LLMs for enumeration questions is remarkable considering that the technology was not designed for this sort of workload (unlike, say, databases). But their performance is far from perfect. Users should exercise caution when using contemporary LLMs to answer enumeration questions, and should not expect consistent responses (even for the same query at different times). Further research is required to understand and address this issue, potentially combining LLMs with other technologies that provide consistency guarantees.

**Limitations and Future Work**   The notion of answer-set (in)consistency could be extended further, for example, to include set cardinality (counts). More work is needed on how to improve the consistency of LLMs in this regard, which may include prompting strategies that instruct the LLM to reason about relations such as containment, disjointness, etc., inherent to such questions. More advanced mitigation strategies could also be explored, such as asking the LLM to parse and reason about structured representations of the questions (perhaps related to work using LLMs to decide SQL equivalence Wei et al. (2025)). Moreover, the experimental setting employed in this study is restricted to isolated, single-turn interactions and does not incorporate multi-turn dialogue. Investigating conversational sequences represents a promising direction for future work, as temporal dependencies may introduce additional inconsistencies or amplify existing ones. The proposed dataset consists of 600 quadruples (2,400 questions) in English and focuses on static, factual domains. While manual curation contributes to the quality and clarity of the question–answer pairs, and the scale of the dataset is sufficient to derive conclusions with statistical significance, future efforts could explore automated or semi-automated approaches for expanding and diversifying the dataset, enabling broader coverage and improved generalizability. Finally, we plan to investigate strategies for mitigating answer-set inconsistency by converting questions into structured representations and subsequently using either the LLM or an external reasoning service to determine the relations between these representations. Overall, one cannot expect an LLM by itself to provide consistency guarantees, so it is of interest to conduct further research on combining LLMs with technologies that provide such guarantees by design.

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

## A   PROMPTS AND PROCESSES USED WITH LLMS

This section presents the three prompting strategies employed to evaluate the LLMs. The place-holder question denotes the text segment that is replaced with a question from the dataset.

### A.1   *Zero-shot*

```
1 {question}
2 If you can't answer, return 'idk'.
3 If the question has no answer, return 'no answer'.
4 In the response, do not use abbreviations or acronyms, but spell out the
      full terms, i.e.  "United States of America" instead of "USA".
5 If the response contains numbers or digits, use Arabic numerals. For
      example, if the answer contains Star Wars V, indicate it with Star
      Wars 5. Do not use Roman numerals (such as V) or text (such as five).
6 Please, Return me an exhaustive list separated by the symbol '|' don't
      add any other text.
```

### A.2   *Classification & Question*

```
1 You are given two questions, q1 and q2.
2 Your task is to determine the logical relationship between their
      respective sets of correct answers
3 Choose only one of the following relations:
4 - Equivalence: The answer sets of q1 and q2 are exactly the same.
5 - Contains: All answers to q2 are also answers to q1, but q1 includes
      additional answers.
6 - ContainedBy: All answers to q1 are also answers to q2, but q2 includes
      additional answers.
7 - Overlap: q1 and q2 share some, but not all, answers. Neither fully
      contains the other.
8 - Disjoint: q1 and q2 have no answers in common.
9 - Unknown: The relation between the answer sets cannot be confidently
      determined based on the given questions.
10 Here are the two questions:
11 q1: {q1}
12 q2: {q2}
13 Return **only** the name of the most appropriate relation from the list
      above.
14 Do **not** provide any explanation or commentary.
```

```
1 You are given three questions: q1, q2, and q3.
2 Each question is associated with a set of answers. Your task is to
      identify the logical relation between the concepts of questions based
      on their answer sets. Compare the relationship of concept between
      the following two sets, s1 and s2:
3  - s1: the set of all answers for q1 that are not answers for q2
4  - s2: the set of answers for q3.
5
6 These are the three questions:
7 q1: {q1}
8 q2: {q2}
9 q3: {q3}
10 For each comparison, use **one of the following labels**:
11 - Equivalence
12 - Contains
13 - ContainedBy
14 - Disjoint
15 - Overlap
16 - Unknown
17 Return only the exactly relation label.
18 Do **not** include any explanation or extra text.
```

### A.3 *Oracle*

Below is the oracle prompt used for the logical equivalence relation. For the other two other logical relations tested, the prompt is adapted by replacing the true_logical_relation placeholder and modifying certain parts of the sentence to ensure coherence.

```
1 Pay attention, the questions I asked you before are {
      true_logical_relation}, but you returned me different values.
2 In the response, do not use abbreviations or acronyms, but spell out the
      full terms, i.e. "United States of America" instead of "USA".
3 If the response contains numbers or digits, use Arabic numerals. For
      example, if the answer contains Star Wars V, indicate it with Star
      Wars 5. Do not use Roman numerals (such as V) or text (such as five).
4 Please, Return me an exhaustive list separated by the symbol '|' don't
      add any other text.
```

### A.4 EVALUATION PIPELINE

After submitting the questions contained in the ASCB using the prompts illustrated previously, with each treated as an independent question, the responses are stored in JSON files where the key corresponds to the question ID and the value is a list of elements representing the set of answers generated by the LLM for that particular question. Once the benchmark has been executed across all models and for all three task types (Base, CtE, and Oracle), all JSON files are provided as input to an evaluation pipeline. This pipeline produces TSV files, one for each evaluated LLM, containing for every answer: the Jaccard similarity, consistency, and a boolean flag indicating whether the answer to the logical relation was satisfied. Finally, as the last step, starting from these task-, model-, and dataset-specific TSV files, four distinct summary TSV files (one for each dataset) are generated. These summary files report the following information: the type of logical relation evaluated, the LLM, the action, the consistency rates, the average non-empty response rate, the average Jaccard similarity, and the ratio of empty responses. All aforementioned TSV file are publicly available in the GitHub repository provided as supplementary material authors (2025).

## B   DATASET CONSTRUCTION

We describe now the dataset construction, consisting of our extraction and modification of questions from the base datasets, followed by a process to filter and identify suitable candidate questions.

### B.1   BASE DATASETS

To construct the question sets, we relied on four data sources: QALD (Usbeck et al., 2024), a handcrafted question-answering dataset for Wikidata and DBpedia; LC-QuAD 2.0, a large-scale question-answering dataset providing SPARQL queries with corresponding answers from both Wikidata and DBpedia[4] ; and a Synthetic dataset, generated entirely through LLMs.

**QALD and LC-QuAD 2.0.** To extract questions from QALD (Liu et al., 2024b; Usbeck et al., 2024) and LC-QuAD 2.0 (Dubey et al., 2019) datasets that conformed to our criteria, we employed a filtering step using an LLM-based pipeline, described in detail in the Section B.2. The resulting questions were reviewed by three authors to ensure adherence to the defined logical relationships and criteria; we remark that careful revisions and considerable adaptations were required by hand, with many candidate questions being discarded. The final filtered QALD and LC-QuAD 2.0 datasets each contain 150 distinct question quadruples (300 for both), for a total of 600 questions in each dataset (1,200 across both).

**QAWiki.** For the QAWiki dataset, many $Q_1$, $Q_2$, and $Q_3$ pairs were directly retrieved via SPARQL queries. However, only 54 $Q_1$–$Q_3$ pairs and 951 $Q_1$–$Q_2$ pairs were available. Therefore, for the 54 $Q_1$–$Q_3$ pairs, the corresponding $Q_2$ and $Q_4$ questions were manually constructed. Conversely, for the 951 $Q_1$–$Q_2$ pairs, $Q_3$ and $Q_4$ were manually derived. As with the other sources,

---

[4]DBpedia: https://www.dbpedia.org/

this dataset was manually curated to include 150 distinct question quadruples per the required relations and criteria, totaling 600 questions.

**Synthetic.** The Synthetic dataset was generated using *gpt-4.1-2025-04-14*, designed to produce 500 complete sets of $Q_1$, $Q_2$, $Q_3$, and $Q_4$ questions. Each set was then manually reviewed by three researchers to ensure correctness, adherence to the inclusion criteria, and to eliminate duplicates. This dataset was heavily curated and revised by hand to include 150 distinct question quadruples, totaling 600 questions. In many cases, the base question tuple generated by the LLM served only as inspiration for the final handcrafted question.

The prompt used to generate the pair of questions $Q_1$ and $Q_2$ is the following:

```
1 Generate {number_of_questions_to_generate} pairs of diverse questions
     about different topics, every pair of questions must be semantically
     equivalent.
2 The answer to every question that you formulate must be a list of values,
      not an ordered list, not a paragraph of text, not a boolean value,
     and not a single number.
3
4 This is an example of a possible pair of questions:
5 1. How many regions of France are there? | How many regions does France
     have?
6 Follow the following format to return the questions:
7 1. Question1 | Equivalent_Question1
8 Do not add any other kind of text except questions.'
```

Following the initial generation of the questions, three of the authors conducted a first round of revision to verify their correctness and semantic equivalence. Based on this refined pool of questions, we then instructed the LLM to generate, for each pair of equivalent questions $(Q_1)$ and $(Q_2)$, a third question $(Q_3)$ whose answer set constitutes a subset of the answers to both $(Q_1)$ and $(Q_2)$. The prompt used for this task is reported below:

```
1 {dataset_of_question_pairs}
2 Starting from the provided dataset of question pairs, where each pair
     consists of two semantically equivalent questions, generate a third
     question whose answer is a subset of the answers to the original two
     questions.
3 The answer to the generated question must be a list of values (not an
     ordered list, not a descriptive paragraph, not a Boolean value, and
     not a single number).
4
5 Use the following output format, and provide only the third question:
6
7 1. Broader_Question_from_the_dataset | Subset_Question
8
9 For example, given the pair: "What countries are in the EU?" | "What
     countries are in the western EU?" the generated question would be the
      subset question.
10
11 Return only the formulated subset question and no additional text.
```

Once the third questions were generated, we manually reviewed each $(Q_3)$ and constructed a corresponding fourth question $(Q_4)$ to satisfy the disjointness relation, defined as $(Q_4 = Q_1 \setminus Q_3)$. The final dataset comprises 600 question quadruples

### B.2 QUESTION PIPELINE

The multi-agent pipeline designed to filter questions that do not satisfy our inclusion criteria was instructed to evaluate each input question against the defined conditions and, upon validation, to generate the corresponding $Q_2$, $Q_3$, and $Q_4$ questions. The LLM employed for this task was GPT-4.1-2025-04-14. The initial datasets consisted of 320 questions from QALD and 30,000 questions from LC-QuAD 2.0. The multi-agent workflow comprises four sequential stages:

- **Agent 1 (Validate Q1):** checks whether the input question $Q_1$ is well-formed and suitable; aborts if invalid.

- **Agent 2 (Generate Q2):** produces a companion question $Q_2$ consistent with the intent and constraints of $Q_1$.

- **Agent 3 (Generate Q3 & Q4):** creates $Q_3$ and $Q_4$ such that the quadruple $(Q_1, Q_2, Q_3, Q_4)$ satisfies the desired logical relations (e.g., equivalence, containment, or disjointness).

- **Agent 4 (Validate All):** verifies that $(Q_1, Q_2, Q_3, Q_4)$ meet the required logical and formatting rules, corrects minor issues, and outputs the final validated set.

The process aborts if any step fails.

Agent1:

```
1    Evaluate whether the following question meets all of the following
     criteria for acceptable answer types:
2
3    1. Returns a limited number of distinct answers (between 2 to 50).
4    2. Does **not** return a binary answer (e.g., "yes" or "no").
5    3. Does **not** return a single specific value (e.g., a date, name,
     or number).
6    4. Does **not** require multiple answer dimensions (e.g., combining "
     what" and "where" in the same question).
7
8    Question: "{question}"
9
10   Answer only with "Yes" or "No".
```

Agent2:

```
1    You are a rephrasing expert. Generate question Q2 which means exactly
      the same thing as the original question but uses different syntax
     wording.
2
3    Original Question: "{question}"
4
5    Format your response as:
6    Q2: ...
```

Agent3:

```
1    Given the original question below, generate Q3 and Q4 to ensure:
2    - Q3 and Q4 are objective,
3    - answer set of Q1 equal to union of answer set of Q3 and Q4,
4    - answer set of Q3 disjoint from Q4,
5    - answer set of Q3 and Q4 subset of Q1 (more restrictive).
6
7    Original Question: "{question}"
8
9    Format your response as:
10   Q3: ...
11   Q4: ...
```

Agent4:

```
1    You are reviewing four related questions.
2
3    Q1: {q1}
4    Q2: {q2}
5    Q3: {q3}
6    Q4: {q4}
7
8    Tasks:
9    1. Check if Q2 is equivalent to Q1.
```

```
10    2. Check if Q3 is a more restrictive version of Q1.
11    3. Check if the union of Q3 and Q4 covers Q1 (Q1 = Q3 + Q4).
12    4. Check all Q1~Q4 are objective questions.
13
14    If everything is correct, answer only:
15    "Yes"
16
17    If not, return a corrected version in this format:
18    Corrected Q1: ...
19    Corrected Q2: ...
20    Corrected Q3: ...
21    Corrected Q4: ...
```

## C  VISUALIZATIONS

In this section, we present visualizations of the main results for answer-set inconsistency across the models.

Figure 1 presents the Jaccard similarity $E_{1,2}^{\text{SIM}}$, $D_{3,4}^{\text{SIM}}$ and $E_{4,1\backslash 3}^{\text{SIM}}$. Figure 1a illustrates that for $E_{1,2}^{\text{SIM}}$ there is high consistency across models, particularly for CtE and Oracle, with Jaccard similarity values frequently exceeding 0.8. In contrast, Figure 1b demonstrates very low consistency for $D_{3,4}^{\text{SIM}}$, indicating that this is the most challenging relation for LLMs. Even when adjusting the prompting strategy, the improvement remains limited and not comparable to the performance observed for $E_{1,2}^{\text{SIM}}$ and $E_{4,1\backslash 3}^{\text{SIM}}$. Finally, similar to $E_{1,2}^{\text{SIM}}$, Figure 1c shows higher consistency for $E_{4,1\backslash 3}^{\text{SIM}}$, with scores improving progressively from Base to Oracle. Moreover, for some models (such as GPT-4.1 and DeepSeek-V3.1), the CtE strategy proves more effective than Oracle.

Figure 2 shows an alternative view of the bar chart, using a line chart and adding the consistency rate. It is interesting to see how the **Base** strategy (blue dot) is almost always below **CtE** and **Ora**. Furthermore, the sorting of the models follows the benchmark (White et al., 2025), but there is no linear increase in performance as the model performance scales (for all strategies) as we expected.

## D  CLASSIFICATION TASK RESULT

Table 5 reports in detail the accuracy of the different models in identifying the different relationships. Across all models, the performance varies from ∼60–91% in terms of average accuracy, depending on the relation. Interestingly, the models perform best, on average, for the disjointness relation ($D_{3,4}$), with an average accuracy across models of ∼60%. The most difficult relationship to recognize for all models was $N_{4,1}$, with an average accuracy of ∼91%. A number of models achieve average accuracy above 90%, including some smaller models, with the best performing (taking ACC: the average across relations) being Gemini-2.5-pro, followed closely by GPT-5, Grok-3-mini, and other models, all of which achieve average accuracy across relations of ∼95%.

## E  STATISTICAL SIGNIFICANCE

Table 6 presents the McNemar $p$-values, which assess whether the **CtE** and **Oracle** strategies significantly outperform the **Base** strategy on the combined dataset. The McNemar test is a paired, non-parametric test that considers only cases where the two methods disagree. In our setting, each case is binary: we check whether the correct relation holds between the given answer sets. Across all quadruples, this results in a total of 2,400 cases.

The table shows that the two proposed strategies offer statistically significant improvements for answer-set consistency on almost all tested models. A statistically-significant improvement occurs in the fewest cases for the $D_{3,4}$ relationship and the **CtE** strategy. The **Oracle** strategy, on the other hand, shows a statistically significant improvement for all cases with only two exceptions: $D_{3,4}$ for the models LLama-3.1-8b and GPT-4.1-nano.

Similarly, we examined whether certain LLMs significantly outperform others to verify hypothesis $H_2$. In this setting, each pair of LLMs is compared, and a corresponding $p$-value indicates the

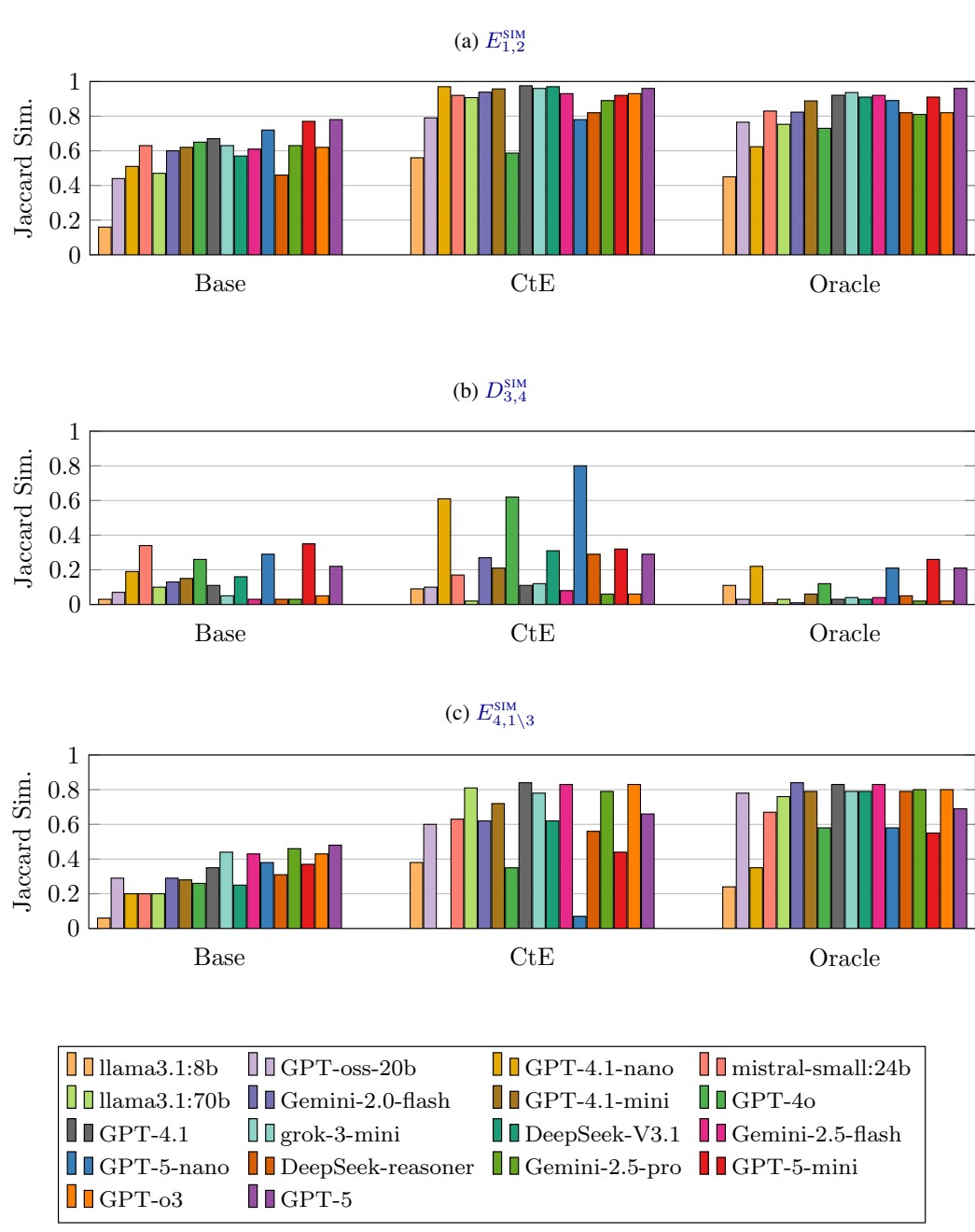

Figure 1: Visualization of answer-set (in)consistency across models

significance level (see Figure 3). This analysis was conducted only for the **Base** strategy, and the results tend to support hypothesis $H_2$.

## F  ANSWER-SET CONTRADICTIONS

In the body of the paper, we have looked at answer-set inconsistencies with respect to gold-standard relations between questions. We further define *answer-set contradictions* as the case where – irrespective of the gold standard – the relation predicted by a model $M$ for questions $Q_i$ and $Q_j$ does not

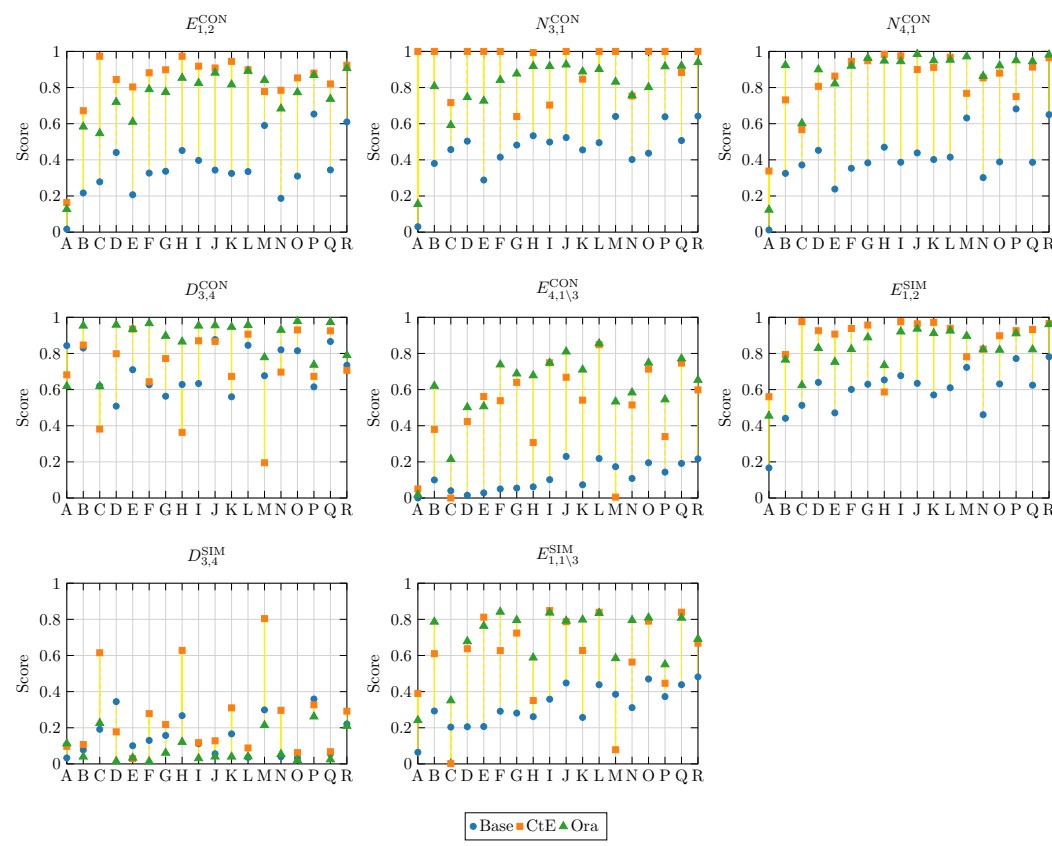

Figure 2: Per-relation consistency for each LLM and strategy. LLMs use IDs from Table 5.

Table 5: Accuracy of different LLMs on the relation classification task across relations. In bold, the model that achieves the highest accuracy for the relation(s). ACC indicates mean accuracy.

| ID | Model | $E_{1,2}^{\text{ACC}}$ | $N_{3,1}^{\text{ACC}}$ | $N_{4,1}^{\text{ACC}}$ | $D_{3,4}^{\text{ACC}}$ | $E_{4,1\backslash3}^{\text{ACC}}$ | ACC |
|---|---|---|---|---|---|---|---|
| A | Llama-3.1-8b | 5.83 | 11.33 | 11.50 | 19.17 | 2.83 | 10.13 |
| B | GPT-oss-20b | 93.83 | 92.33 | 96.33 | 91.33 | 94.00 | 93.56 |
| C | GPT-4.1-nano | 69.17 | 39.33 | 41.33 | 62.33 | 0.00 | 42.43 |
| D | Mistral-small:24b | 91.69 | 0.17 | 1.19 | 99.49 | 11.86 | 40.88 |
| E | Llama-3.1-70b | **97.33** | 7.67 | 27.00 | 99.17 | 98.50 | 65.93 |
| F | Gemini-2.0-flash | 90.83 | 79.17 | 21.33 | 95.17 | 80.33 | 73.37 |
| G | GPT-4.1-mini | 91.00 | 34.50 | 20.83 | 99.00 | 95.83 | 68.23 |
| H | GPT-4o | 94.50 | 31.67 | 12.17 | 99.33 | 65.33 | 60.60 |
| I | GPT-4.1 | 94.50 | **98.67** | 69.67 | **99.50** | **99.33** | 92.33 |
| J | Grok-3-mini | 91.67 | 94.83 | 96.50 | 97.67 | 96.83 | 95.50 |
| K | DeepSeek-V3.1 | 96.00 | 35.50 | 20.83 | 99.33 | 90.50 | 68.43 |
| L | Gemini-2.5-flash | 86.33 | 95.67 | 96.33 | 96.83 | 95.50 | 94.13 |
| M | GPT-5-nano | 91.83 | 85.00 | 91.50 | 88.50 | 91.17 | 89.60 |
| N | DeepSeek-reasoner | 88.83 | 97.00 | 96.67 | 96.33 | 94.33 | 94.63 |
| O | Gemini-2.5-pro | 91.33 | 97.67 | 94.50 | 98.50 | 96.17 | **95.63** |
| P | GPT-5-mini | 90.33 | 97.17 | 97.83 | 95.00 | 95.17 | 95.10 |
| Q | GPT-o3 | 92.71 | 97.12 | 97.63 | 95.25 | 93.73 | 95.29 |
| R | GPT-5 | 90.50 | 97.67 | **98.33** | 96.83 | 94.50 | 95.57 |
| *Average* | | 86.01 | 66.25 | 60.64 | **90.48** | 77.55 | 76.19 |

hold for the answer sets that $M$ itself generates for $Q_i$ and $Q_j$. In these cases, the model contradicts itself. We describe here some measures and results that we extracted to analyze this issue.

## F.1 MEASURES

We consider the following measures to quantify the self-contradictions of the LLMs in this setting.

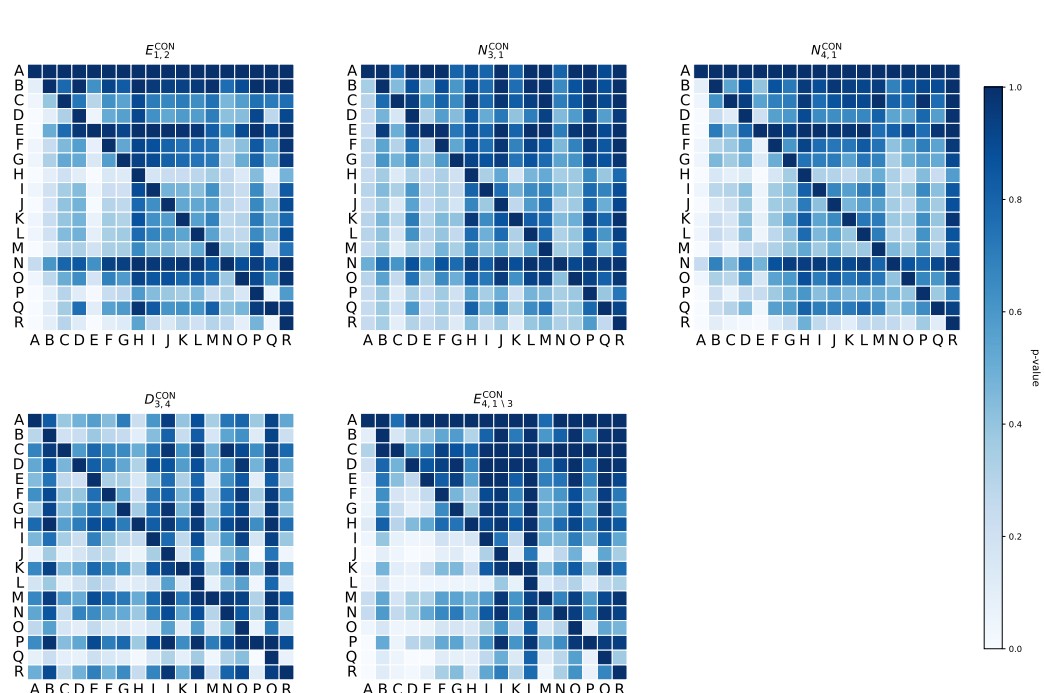

Figure 3: $p$-value heatmap of LLMs in overall datasets for Base strategy.

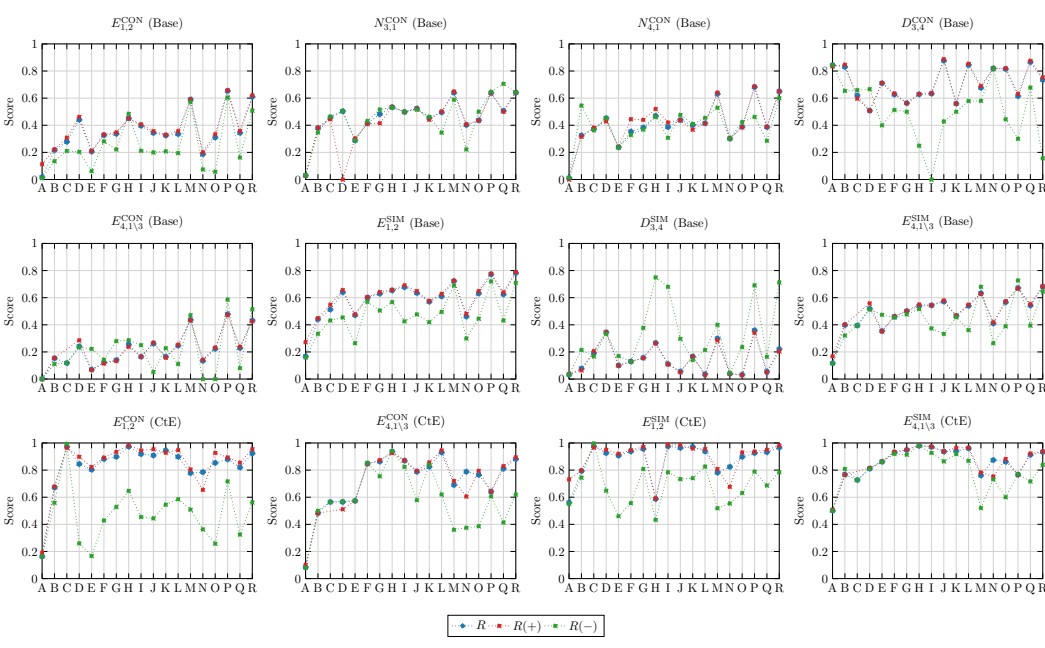

Figure 4: Comparison of R and R(+/−) consistency for Base (1st and 2nd rows) and CtE (3rd row).

Table 6: McNemar $p$-value which shows the statistically significant improvements over the zero-shot prompting with the Classification-then-Enumerate (CtE) strategy and Oracle (Ora.) on the overall dataset. Asterisks denote statistical significance: *** for $p < 0.001$, ** for $p < 0.01$, and * for $p < 0.05$. The empty cell indicates a p-value $\geq 0.05$

| LLM | CtE | | | | | Ora. | | | | |
|---|---|---|---|---|---|---|---|---|---|---|
| | $E_{1,2}$ | $N_{3,1}$ | $N_{4,1}$ | $D_{3,4}$ | $E_{4,1\backslash 3}$ | $E_{1,2}$ | $N_{3,1}$ | $N_{4,1}$ | $D_{3,4}$ | $E_{4,1\backslash 3}$ |
| Llama-3.1-8b | *** | *** | *** | | *** | *** | *** | *** | | *** |
| GPT-oss-20b | *** | *** | *** | | *** | *** | *** | *** | *** | *** |
| GPT-4.1-nano | *** | *** | *** | | *** | *** | *** | *** | | *** |
| Mistral-small:24b | *** | *** | *** | *** | *** | *** | *** | *** | *** | *** |
| Llama-3.1-70b | *** | *** | *** | *** | *** | *** | *** | *** | *** | *** |
| Gemini-2.0-flash | *** | *** | *** | | *** | *** | *** | *** | *** | *** |
| GPT-4.1-mini | *** | *** | *** | *** | *** | *** | *** | *** | *** | *** |
| GPT-4o | *** | *** | *** | | *** | *** | *** | *** | *** | *** |
| GPT-4.1 | *** | *** | *** | *** | *** | *** | *** | *** | *** | *** |
| Grok-3-mini | *** | *** | *** | | *** | *** | *** | *** | *** | *** |
| DeepSeek-V3.1 | *** | *** | *** | *** | *** | *** | *** | *** | *** | *** |
| Gemini-2.5-flash | *** | *** | *** | *** | *** | *** | *** | *** | *** | *** |
| GPT-5-nano | *** | *** | *** | | | *** | *** | *** | *** | *** |
| DeepSeek-reasoner | *** | *** | *** | | *** | *** | *** | *** | *** | *** |
| Gemini-2.5-pro | *** | *** | *** | *** | *** | *** | *** | *** | *** | *** |
| GPT-5-mini | *** | *** | *** | *** | *** | *** | *** | *** | *** | *** |
| GPT-o3 | *** | *** | *** | *** | *** | *** | *** | *** | *** | *** |
| GPT-5 | *** | *** | *** | | *** | *** | *** | *** | *** | *** |

**Contradiction-free rates**   To complement classification accuracy and answer-set consistency, we measure whether a model is *internally consistent* with respect to the logical relation it *predicts*. Given two questions $Q_i, Q_j$, We say the case is *contradiction-free* iff the predicted relation is satisfied by the model's *own* answers; otherwise, a *self-contradiction* occurs. We define the *self-contradiction rate* as the percentage of cases where a model's predicted relation is not satisfied by its own answers. This measure distinguishes between models that are internally consistent but wrong and those that are internally inconsistent, thereby offering a finer-grained perspective on model reliability. We denote such rates for a model $M$ and relation $R$ as $R^{\text{CFR}}(M)$.

**Consistency by relation correctness**   To further analyze model behavior, we distinguish consistency depending on whether the logical relation between two questions is correctly identified with respect to the gold standard. This breakdown reveals how much consistency stems from correctly recognizing the relation versus how much persists even when the relation is misclassified.

### F.2   RESULTS

In Table 7, we present the results of the contradiction-free rates for five relations. We see that the models do tend to contradict themselves, i.e., the answers they return for questions do not respect the relation between the questions that they themselves predict. A lot of variance is seen across the models, with large models showing more consistency.

Furthermore, we examine how consistency varies depending on whether the relation classification is correct. Table 8 presents the results for consistency and similarity when the predicted relation is correct, while Table 9 shows the corresponding results when the relation is incorrect. Figure 4 compares consistency across all cases, as well as positive and negative subsets. The results indicate that incorrect relation classification generally leads to lower consistency.

## G   CAUSAL ANALYSIS OF LLMS INCONSISTENCY

As we observed, LLMs often give varying responses to identical inputs, potentially due to four factors: decoding randomness, computational nondeterminism, order sensitivity, and data-level conflicts.

Table 7: Per-relation contradiction-free rates $R^{\text{CFR}}$ (0–100 scale).

| LLM | Str | $E_{1,2}^{\text{CFR}}$ | $N_{3,1}^{\text{CFR}}$ | $N_{4,1}^{\text{CFR}}$ | $D_{3,4}^{\text{CFR}}$ | $E_{4,1\backslash3}^{\text{CFR}}$ |
|---|---|---|---|---|---|---|
| Llama-3.1-8b | Base | 93.83 | 86.33 | 87.33 | 28.50 | 97.17 |
| | CtE | 81.00 | 13.83 | –"– | –"– | 82.67 |
| GPT-oss-20b | Base | 26.17 | 40.33 | 32.17 | 80.33 | 16.00 |
| | CtE | 66.50 | 89.33 | –"– | –"– | 40.33 |
| GPT-4.1-nano | Base | 45.67 | 50.00 | 53.17 | 50.00 | 96.00 |
| | CtE | 69.33 | 36.50 | –"– | –"– | 100.00 |
| Mistral-small:24b | Base | 48.98 | 49.49 | 54.58 | 50.68 | 87.29 |
| | CtE | 88.50 | 0.17 | –"– | –"– | 56.50 |
| Llama-3.1-70b | Base | 23.00 | 68.17 | 62.17 | 71.17 | 4.33 |
| | CtE | 82.33 | 9.00 | –"– | –"– | 53.67 |
| Gemini-2.0-flash | Base | 37.33 | 44.50 | 62.33 | 62.50 | 23.33 |
| | CtE | 88.50 | 46.83 | –"– | –"– | 46.17 |
| GPT-4.1-mini | Base | 38.67 | 46.00 | 59.17 | 56.33 | 8.67 |
| | CtE | 89.33 | 44.83 | –"– | –"– | 62.50 |
| GPT-4o | Base | 45.33 | 49.00 | 53.50 | 63.17 | 37.67 |
| | CtE | 96.50 | 66.33 | –"– | –"– | 73.50 |
| GPT-4.1 | Base | 42.83 | 49.83 | 50.33 | 63.83 | 10.83 |
| | CtE | 92.33 | 69.67 | –"– | –"– | 74.67 |
| Grok-3-mini | Base | 39.33 | 52.17 | 44.00 | 88.00 | 26.17 |
| | CtE | 91.83 | 95.17 | –"– | –"– | 69.33 |
| DeepSeek-V3.1 | Base | 34.83 | 50.33 | 54.33 | 56.00 | 14.83 |
| | CtE | 89.33 | 10.00 | –"– | –"– | 52.00 |
| Gemini-2.5-flash | Base | 41.83 | 50.83 | 41.83 | 84.00 | 25.67 |
| | CtE | 87.50 | 94.83 | –"– | –"– | 85.33 |
| GPT-5-nano | Base | 57.83 | 61.33 | 62.67 | 65.83 | 24.50 |
| | CtE | 77.67 | 89.33 | –"– | –"– | 8.83 |
| DeepSeek-reasoner | Base | 28.17 | 41.83 | 31.50 | 79.67 | 16.50 |
| | CtE | 65.33 | 92.00 | –"– | –"– | 54.00 |
| Gemini-2.5-pro | Base | 38.67 | 43.67 | 39.67 | 81.67 | 23.33 |
| | CtE | 90.67 | 95.83 | –"– | –"– | 74.67 |
| GPT-5-mini | Base | 63.33 | 63.00 | 68.33 | 63.50 | 19.17 |
| | CtE | 84.67 | 96.17 | –"– | –"– | 38.00 |
| GPT-o3 | Base | 39.32 | 49.49 | 39.66 | 84.92 | 25.42 |
| | CtE | 84.33 | 85.67 | –"– | –"– | 77.50 |
| GPT-5 | Base | 60.83 | 63.50 | 64.67 | 75.67 | 26.83 |
| | CtE | 91.33 | 98.17 | –"– | –"– | 63.00 |

## G.1 DECODING RANDOMNESS

The most immediate cause of nondeterminism in LLM output is the probabilistic nature of token generation. The sampling temperature $T$ (Ackley et al., 1985; Li et al., 2025) acts as a hyperparameter that influences the output distribution. Formally, the probability of selecting token $w_i$ is: $P(w_i) = \frac{e^{z_i/T}}{\sum_j e^{z_j/T}}$, where $z_i$ denotes the logit for token $w_i$. When $T > 0$, sampling methods such as top-$k$ and nucleus (top-$p$) sampling apply stochasticity to improve diversity, ensuring that identical inputs can yield semantically distinct outputs. Theoretically, setting $T = 0$ (greedy decoding) should guarantee determinism. However, empirical studies (Li et al., 2025; Renze, 2024; Wang & Wang, 2025; Atil et al., 2024) demonstrate that outputs often remain unstable even with the temperature set to zero, pointing to other underlying causes of nondeterminism.

## G.2 COMPUTATIONAL NONDETERMINISM

Nondeterminism at $T = 0$ can arise from the non-associative nature of floating-point arithmetic in GPUs, where $(a + b) + c \neq a + (b + c)$ due to rounding errors. As Yuan et al. demonstrates, minor numerical discrepancies in early tokens can cascade into divergent outputs, particularly in reasoning-heavy tasks. Furthermore, modern optimizations, such as Mixture-of-Experts (MoE) (Masoudnia & Ebrahimpour, 2014), FlashAttention (Dao et al., 2022), and continuous batching often process data

Table 8: Per-relation consistency rate and similarity when predicted relation is correct according to ground-truth (0–100 scale)

| LLM | Act | $E^{\text{CON}}_{1,2}$ | $N^{\text{CON}}_{3,1}$ | $N^{\text{CON}}_{4,1}$ | $D^{\text{CON}}_{3,4}$ | $E^{\text{CON}}_{4,1\backslash3}$ | $E^{\text{SIM}}_{1,2}$ | $D^{\text{SIM}}_{3,4}$ | $E^{\text{SIM}}_{4,1\backslash3}$ |
|---|---|---|---|---|---|---|---|---|---|
| Llama-3.1-8b | Base | 11.43 | 2.94 | 0.00 | 83.48 | 0.00 | 27.31 | 3.64 | 16.82 |
|  | CtE | 19.23 | 100.00 | –"– | –"– | 10.26 | 73.28 | –"– | 50.77 |
| GPT-oss-20b | Base | 22.20 | 38.27 | 31.66 | 84.67 | 15.60 | 44.83 | 6.45 | 40.29 |
|  | CtE | 67.84 | 100.00 | –"– | –"– | 48.26 | 79.69 | –"– | 76.58 |
| GPT-4.1-nano | Base | 30.84 | 44.49 | 38.31 | 59.63 | –"– | 54.88 | 20.64 | –"– |
|  | CtE | 96.51 | 66.23 | –"– | –"– | –"– | 96.72 | –"– | –"– |
| Mistral-small:24b | Base | 46.21 | 0.00 | 42.86 | 50.77 | 28.57 | 65.68 | 34.47 | 55.83 |
|  | CtE | 89.82 | 100.00 | –"– | –"– | 51.16 | 95.08 | –"– | 81.67 |
| Llama-3.1-70b | Base | 21.06 | 30.43 | 24.07 | 71.26 | 6.77 | 47.69 | 10.00 | 35.24 |
|  | CtE | 82.30 | 100.00 | –"– | –"– | 57.27 | 92.09 | –"– | 86.26 |
| Gemini-2.0-flash | Base | 33.21 | 41.03 | 44.53 | 63.46 | 11.60 | 60.45 | 13.01 | 46.20 |
|  | CtE | 89.25 | 100.00 | –"– | –"– | 84.55 | 94.75 | –"– | 93.57 |
| GPT-4.1-mini | Base | 34.80 | 41.55 | 44.00 | 56.40 | 13.39 | 64.19 | 15.52 | 50.27 |
|  | CtE | 93.42 | 65.32 | –"– | –"– | 87.39 | 97.17 | –"– | 95.15 |
| GPT-4o | Base | 44.97 | 53.68 | 52.05 | 63.09 | 23.79 | 65.84 | 26.36 | 55.03 |
|  | CtE | 98.28 | 99.25 | –"– | –"– | 92.57 | 59.12 | –"– | 98.35 |
| GPT-4.1 | Base | 40.74 | 49.83 | 42.11 | 63.65 | 16.44 | 69.17 | 10.90 | 54.46 |
|  | CtE | 94.53 | 70.34 | –"– | –"– | 87.14 | 98.65 | –"– | 97.24 |
| Grok-3-mini | Base | 35.64 | 52.37 | 43.70 | 88.74 | 26.85 | 64.90 | 5.04 | 57.93 |
|  | CtE | 95.42 | 100.00 | –"– | –"– | 79.52 | 98.67 | –"– | 93.89 |
| DeepSeek-V3.1 | Base | 32.99 | 44.13 | 36.80 | 56.04 | 15.84 | 57.66 | 16.63 | 46.73 |
|  | CtE | 92.81 | 100.00 | –"– | –"– | 85.71 | 95.98 | –"– | 96.43 |
| Gemini-2.5-flash | Base | 35.71 | 50.17 | 41.35 | 85.37 | 25.31 | 62.83 | 3.01 | 54.82 |
|  | CtE | 94.79 | 100.00 | –"– | –"– | 94.57 | 95.60 | –"– | 96.53 |
| GPT-5-nano | Base | 59.17 | 64.90 | 64.12 | 68.93 | 43.33 | 72.58 | 28.56 | 62.84 |
|  | CtE | 80.55 | 100.00 | –"– | –"– | 72.00 | 80.81 | –"– | 78.26 |
| DeepSeek-reasoner | Base | 20.08 | 40.72 | 30.17 | 82.01 | 14.31 | 48.13 | 3.94 | 41.90 |
|  | CtE | 65.47 | 100.00 | –"– | –"– | 60.56 | 67.65 | –"– | 75.33 |
| Gemini-2.5-pro | Base | 33.39 | 43.52 | 38.62 | 82.06 | 23.22 | 64.90 | 2.95 | 57.36 |
|  | CtE | 92.70 | 100.00 | –"– | –"– | 79.50 | 93.07 | –"– | 88.22 |
| GPT-5-mini | Base | 65.87 | 63.81 | 68.65 | 63.16 | 47.29 | 77.73 | 34.18 | 66.80 |
|  | CtE | 89.35 | 100.00 | –"– | –"– | 64.34 | 93.71 | –"– | 76.66 |
| GPT-o3 | Base | 35.83 | 50.09 | 38.89 | 87.54 | 23.87 | 63.97 | 4.98 | 55.25 |
|  | CtE | 85.54 | 88.41 | –"– | –"– | 83.01 | 94.99 | –"– | 92.23 |
| GPT-5 | Base | 62.06 | 64.16 | 65.08 | 75.39 | 42.68 | 78.91 | 20.50 | 68.34 |
|  | CtE | 95.47 | 100.00 | –"– | –"– | 89.49 | 98.21 | –"– | 93.82 |

in non-deterministic orders to maximize throughput. Atil et al. (2024) highlight that while fixing random seeds can mitigate multi-GPU randomness, engineering optimizations (e.g., chunk prefilling or prefix caching) frequently reintroduce nondeterministic behavior, making exact reproducibility fragile even in controlled environments.

## G.3 ORDER SENSITIVITY

LLMs exhibit hypersensitivity to input presentation due to the self-attention mechanism (Vaswani et al., 2017). Because attention weights rely on positional encodings, trivial formatting changes – such as reordering few-shot examples or altering whitespace – can have notable effects on output. Liu et al. (2024a) identifies a "lost in the middle" phenomenon, where models fail to access information located in the middle of a long context window. Similarly, Lu et al. (2022) shows that the chosen order of few-shot examples can change performance from near-random to state-of-the-art.

## G.4 DATA-LEVEL CONFLICTS

Nondeterminism may also arise due to differences between pre-trained weights (parametric memory) and the provided context (non-parametric memory). Xu et al. (2024) identifies knowledge

Table 9: Per-relation consistency rate and similarity when predicted relation is incorrect according to ground-truth (0–100 scale)

| LLM | Act | $E^{CON}_{1,2}$ | $N^{CON}_{3,1}$ | $N^{CON}_{4,1}$ | $D^{CON}_{3,4}$ | $E^{CON}_{4,1\backslash3}$ | $E^{SIM}_{1,2}$ | $D^{SIM}_{3,4}$ | $E^{SIM}_{4,1\backslash3}$ |
|---|---|---|---|---|---|---|---|---|---|
| Llama-3.1-8b | Base | 1.06 | 3.01 | 1.32 | 84.54 | 0.17 | 16.01 | 3.21 | 11.58 |
|  | CtE | 16.20 | 100.00 | –"– | –"– | 8.05 | 55.25 | –"– | 50.09 |
| GPT-oss-20b | Base | 13.51 | 34.78 | 54.55 | 65.38 | 11.11 | 33.38 | 21.46 | 32.12 |
|  | CtE | 55.88 | 100.00 | –"– | –"– | 50.00 | 74.50 | –"– | 80.90 |
| GPT-4.1-nano | Base | 21.08 | 46.43 | 36.36 | 65.93 | 11.83 | 43.23 | 16.67 | 39.49 |
|  | CtE | 99.41 | 73.50 | –"– | –"– | 56.50 | 99.71 | –"– | 72.72 |
| Mistral-small:24b | Base | 20.41 | 50.42 | 45.28 | 66.67 | 23.46 | 45.42 | 33.33 | 51.40 |
|  | CtE | 26.00 | 100.00 | –"– | –"– | 56.91 | 64.90 | –"– | 81.06 |
| Llama-3.1-70b | Base | 6.25 | 28.70 | 23.74 | 40.00 | 22.22 | 26.46 | 17.00 | 47.39 |
|  | CtE | 16.67 | 100.00 | –"– | –"– | 57.52 | 46.07 | –"– | 86.09 |
| Gemini-2.0-flash | Base | 28.12 | 43.18 | 32.84 | 51.28 | 14.29 | 56.97 | 12.86 | 45.45 |
|  | CtE | 42.86 | 100.00 | –"– | –"– | 85.25 | 55.66 | –"– | 91.98 |
| GPT-4.1-mini | Base | 22.22 | 51.65 | 36.84 | 50.00 | 28.00 | 50.59 | 37.72 | 47.75 |
|  | CtE | 52.83 | 63.47 | –"– | –"– | 75.47 | 80.86 | –"– | 91.40 |
| GPT-4o | Base | 48.48 | 53.17 | 46.30 | 25.00 | 28.71 | 56.77 | 75.00 | 51.64 |
|  | CtE | 64.71 | 99.50 | –"– | –"– | 94.12 | 43.37 | –"– | 97.87 |
| GPT-4.1 | Base | 21.21 | 50.00 | 30.77 | 0.00 | 25.00 | 42.59 | 68.06 | 37.38 |
|  | CtE | 45.45 | 70.00 | –"– | –"– | 82.35 | 78.50 | –"– | 92.73 |
| Grok-3-mini | Base | 20.00 | 51.61 | 47.62 | 42.86 | 5.26 | 47.73 | 29.75 | 33.28 |
|  | CtE | 44.44 | 100.00 | –"– | –"– | 57.89 | 73.36 | –"– | 86.46 |
| DeepSeek-V3.1 | Base | 20.83 | 46.25 | 41.05 | 50.00 | 22.81 | 41.98 | 14.06 | 45.65 |
|  | CtE | 54.55 | 100.00 | –"– | –"– | 83.92 | 74.09 | –"– | 91.83 |
| Gemini-2.5-flash | Base | 19.51 | 34.62 | 45.45 | 57.89 | 11.11 | 49.53 | 21.45 | 36.10 |
|  | CtE | 58.54 | 100.00 | –"– | –"– | 62.07 | 82.66 | –"– | 86.86 |
| GPT-5-nano | Base | 57.14 | 58.89 | 52.94 | 57.97 | 47.17 | 68.94 | 40.02 | 68.06 |
|  | CtE | 50.91 | 100.00 | –"– | –"– | 36.00 | 51.92 | –"– | 52.07 |
| DeepSeek-reasoner | Base | 7.46 | 22.22 | 30.00 | 81.82 | 0.00 | 29.98 | 4.50 | 26.44 |
|  | CtE | 36.36 | 100.00 | –"– | –"– | 37.50 | 55.45 | –"– | 73.20 |
| Gemini-2.5-pro | Base | 5.77 | 50.00 | 42.42 | 44.44 | 0.00 | 44.58 | 23.57 | 39.00 |
|  | CtE | 25.76 | 100.00 | –"– | –"– | 38.64 | 63.17 | –"– | 60.09 |
| GPT-5-mini | Base | 60.34 | 64.71 | 46.15 | 30.00 | 58.62 | 72.10 | 69.17 | 72.76 |
|  | CtE | 71.74 | 100.00 | –"– | –"– | 60.71 | 78.90 | –"– | 77.00 |
| GPT-o3 | Base | 16.28 | 70.59 | 28.57 | 67.86 | 8.11 | 43.24 | 16.46 | 39.48 |
|  | CtE | 32.50 | 86.36 | –"– | –"– | 41.38 | 68.66 | –"– | 71.69 |
| GPT-5 | Base | 50.88 | 64.29 | 60.00 | 15.79 | 51.52 | 71.02 | 71.31 | 64.70 |
|  | CtE | 56.25 | 100.00 | –"– | –"– | 62.07 | 78.48 | –"– | 84.04 |

conflicts, where externally retrieved context contradicts internal training data, as a source of nondeterminism. Another issue is due to temporal misalignment; Nakshatri et al. (2025) finds that while models may contain updated knowledge, they often struggle to retrieve it reliably against outdated but strongly-weighted internal facts. Consequently, models may switch in an unpredictable manner between "local optima," prioritizing helpfulness over correctness (Xie et al., 2023).

# H  INCONSISTENCY ERROR ANALYSIS

We conducted a detailed analysis of the errors that continue to cause inconsistencies in the LLMs responses, even when applying the CtE and Oracle mitigation strategies. We identified the following recurring error patterns, which can be classified into the categories listed below, excluding those errors attributable to the stochastic nature of the model, which have already been discussed in Appendix G.

- Use of different terminology to refer to the same concept or entity: In some questions involving country lists, the model refers to the same country using different formulations, for instance, "Spain" versus "Kingdom of Spain". This is despite the fact that our prompting guides the LLM to provide full names, avoid abbreviations, etc., to minimize such cases.

- Completeness of the response: For the case of relation $E_{1,2}$, we observed instances in which the same question produced answers with different set cardinalities across different executions.

- Incapability of capturing implicit logic: As discussed in Appendix F.2, LLMs that misclassify the relation often show lower consistency in subsequent answers. When the relation is incorrect, follow-up reasoning becomes unreliable. This highlights that the ability to correctly grasp the implicit logic and contextual relationships behind the questions is crucial for achieving higher consistency.

- Lack option for uncertainty: The strategy of permitting LLMs to answer "IDK" when uncertain generally improves consistency. Without this option, LLMs may attempt to answer despite uncertainty, often leading to hallucinated responses that cause inconsistency.

## I  GENAI USAGE DISCLOSURE

GenAI is used for text refinement, assist code debugging, dataset creation.

