# OpenReview forum: "Answer-Set Consistency of LLMs for Question Answering"
_ICLR.cc/2026/Conference — Submitted to ICLR 2026_

### Official Review · Reviewer_ykMU · 2025-10-28

**Soundness:** 3
**Presentation:** 3
**Contribution:** 2
**Rating:** 4
**Confidence:** 3

**Summary:**

This paper formalizes the problem of answer-set inconsistency in large language models (LLMs) for factual enumeration questions,
where LLMs generate responses that violate set-theoretic relations (e.g., equivalence, containment, disjoint-ness) between questions. The paper proposes a benchmark dataset (ASCB) with 600 handcrafted question quadruples (2,400 questions) and evaluate 18 state-of-the-art LLMs. Key contributions include a novel theoretical framework, comprehensive metrics, and mitigation strategies that significantly improve consistency. Experiments demonstrate pervasive inconsistency across models, even when they correctly recognize relations, with mitigation
strategies achieving statistical significance (p < 0.001).

**Strengths:**

S1. The paper introduces a rigorous theoretical foundation for answer-set consistency, formalizing it using set-theoretic relations
(Section 3.1). This includes definitions for consistency and contradictions, enabling a sampling-based evaluation approach
with error guarantees (e.g., control relation R* for stochasticity analysis in Section 3.4). The framework bridges LLM behavior with
database-like query containment principles, advancing beyond prior work focused on single-answer consistency.

S2. The empirical analysis is extensive, covering 18 LLMs (e.g., DeepSeek, Grok, Mistral, GPT, Gemini, Llama families) across
multiple tasks (Base evaluation, CtE, Oracle). Evaluation metrics (Consistency rates, Jaccard similarity) and statistical testing
(McNemar test) reveal pervasive inconsistency.

S3. The mitigation strategies (e.g., CtE) of inconsistency are practical and only require minimal prompt engineering, making
them accessible for real-world applications.

**Weaknesses:**

W1. The proposed ASCB dataset is limited to 600 quadruples (2,400 questions) in English, focusing on static, factual domains.
This scale may not generalize to dynamic environments or high stakes applications requiring larger scales datasets. The manual
curation, while ensuring quality, restricts diversity and scalability.

W2. Based on KGQA datasets, the experiments are conducted in isolated, single-turn contexts, ignoring real-world scenarios like
multi-turn dialogues. This limits the validity of consistency claims for interactive systems, where temporal dynamics might
exacerbate inconsistencies.

W3. While the paper highlights stochasticity as a cause of inconsistency in Section 3.4 and 4.2, it lacks accurate error reason analysis (e.g., entity-level semantic misunderstanding or knowledge gaps). For example, contradictions in Appendix F (Table 7) are not decomposed into specific error types, lacking of guidance to extend datasets and hindering targeted model improvements based on prompt strategy.

W4. The paper describes different relations (including primary and implied relations) in Table 2. However, there is an unclear
explanation about modeling 12 pairs of questions in Section 3.2 and even why paper chooses five primary relations (R1-R5) in
Section 3.3.

**Questions:**

Limitations in dataset diversity, dynamic scenario evaluation, and error analysis reduce its generalizability. See the weaknesses (W1-W4) above.

---

> ### Author Response · Authors · 2025-11-21
> **Reply to Reviewer ykMU for W1-W4**
>
> We are grateful for your valuable comments and time, and for this opportunity to improve our work. We respond to the weaknesses indicated.
>
> W1: We appreciate that our work is currently limited in terms of providing insights into dynamic environments. Our work introduces the answer-set consistency problem, and provides a dataset of hand-crafted English questions to address the static case. Our dataset of 600 question tuples (2,400 questions in total), is sufficient to draw various conclusions with statistical significance regarding the static case of general knowledge questions. We believe that our work thus provides a solid platform from which to explore answer-set (in)consistency of LLMs in other key settings, as you mentioned. We’ve extended our future work section along these lines.
>
> W2: Similarly, we agree that studying answer-set consistency in multi-turn dialogues would be very interesting: though not in our current scope, our contributions would facilitate such analysis in future. We have extended our future work section along these lines.
>
> W3: We appreciate the comments, and we are now working on analyzing error cases, where we present additional discussion in Appendices G and H, and we’ve also extended our discussion on results in Section 4.2 along these lines. Based on a preliminary analysis of the errors, the main issues we have identified are: (i) the use of different terminology to refer to the same concept or entity [which we address by requesting models to return full names, avoid abbreviations, etc.], (ii) incompleteness caused by knowledge gaps, and (iii) stochasticity. Other specific cases involve models being unable to classify relations, or misinterpreting prompts (per the cases now discussed in Section 4.2).
>
> W4: Our initial focus was on testing equivalence and containment, which are the classical decision problems studied for structured queries (e.g., SQL queries) in the database literature. When generating the questions for these, we realized that we could add (with relatively low manual effort) another question to test another case of containment as well as disjointness (see response to Reviewer Cckv on the rationale for building our dataset from quadruples). Other relations between relations could be considered, where in future work we mention cardinality-based consistency: does the number of answers generated for an enumeration question (e.g., “What countries are in the E.U.?”) correspond to the answer generated for the count version of the question (e.g., “How many countries are in the E.U.?”). However, the relations considered in the current submission already cover, and go beyond, the classical relations considered for static analysis of structured queries like SQL. In terms of selecting five relations, we avoid symmetric and inverse relations as these cases would be repetitive, creating dependencies between results; for example, asking if question $Q_1$ is broader than question $Q_3$ and later asking if question $Q_3$ is narrower than $Q_1$ would be redundant. This yields five relations. We extended the description of the dataset construction in Section 3.2 to clarify these design choices, and the description on the choice of the 5 relations in Section 3.3 (Questions and relations considered).
>
> (Please note that we have uploaded a revised version of our submission for the reviewers’ consideration, with changes to address reviewer comments highlighted in blue.)
>
> **Thank you again for your valuable feedback! Since the rebuttal window closes in just a few days, it would be great if we could have a conversational discussion. We truly appreciate your time and look forward to your response!**

---

> > ### Comment · Reviewer_ykMU · 2025-11-28
> >
> > Thank you for the feedback. My concerns on W1, W2 remain and on W3 is partially addressed. I'll retain my score.

---

### Official Review · Reviewer_Cckv · 2025-10-31

**Soundness:** 3
**Presentation:** 1
**Contribution:** 2
**Rating:** 2
**Confidence:** 4

**Summary:**

The paper is concerned with answer-set consistency of LLMs during question answering. When asked an *enumerative* question, the LLMs answers with a set. LLMs may contradict themselves when enumerating all entities satisfying the question. To this end, the authors consider consistency evaluation from a set-theoretic perspective, by checking equivalence, disjointness, containment of set operations, etc.


The paper proposes a benchmark, and reveal inconsistencies of existing LLMs on answer-set setting.

**Strengths:**

The setup is well motivated, since LLMs may be inconsistent when asked to enumerate all entities in a set-focused question. The related work is explained well, with prior studies' definitions of inconsistencies. Empirical results are extensive, covering many models, and showing that they are not yet fully consistent during question answering. Several parts of the paper can be improved, as explained below.

**Weaknesses:**

- Section 3 requires rewriting. Mathematical notations could be simplified (e.g., table 2 should be as simple as possible). Some notations do not look standard, or not explained well. Since the authors consider a finite number of set operators, it is better to use in a *case by case* what it means to achieve consistency under equivalence, containment, disjointness or overlap. The notations for disjointness and overlap are mixed with Boolean True or False -- I have never seen these notations before. In summary, the consistency criteria should be explained in an intuitive manner, with illustrative examples.

- There is no discussion on what prompts are given to the LLM and how answers are parsed -- they have major influence on the evaluation of inconsistency.

- The mitigation strategy has a *conversational* aspect, which may improve consistency in a surface level, i.e., in-context, and not to the parametric knowledge of the LLM. What happens when the LLM does not know the enumerative answer?

- The construction of the dataset is explained in a complicated manner. Why there are quadruples in line 202, and onwards?

- Several sentences are repeated multiple times, but demonstrative examples are missing. Examples are line 182 to 185. Put differently, since the objective is to improve consistency, it is better to show examples of consistency/inconsistency without being too abstract in notations (refer to Ghosh et at (2025) for inspiration).

**Questions:**

Please address the points mentioned in the weakness.

I am willing to increase score after rebuttal.

---

> ### Author Response · Authors · 2025-11-21
> **Reply to Reviewer 3 Cckv for weaknesses**
>
> Thanks a lot for your valuable comments and time, and this opportunity to improve our work. We respond to the weaknesses as requested.
>
> Section 3 requires rewriting: We apologize for the unclear notation. We have significantly rewritten Section 3 to provide a simpler notation as suggested, removing non-standard notation. We have also added Table 2 with an explanation of the notation we introduce for relations. Other sections have likewise been revised to use the simplified notation.There are also some concrete cases added.
>
> LLM prompts: We present prompts in Appendix A for the experiments, and Appendix B for dataset construction and refinement (due to space reasons we cannot fit them in the body of the paper; full scripts and documentation are also available in the anonymous repository linked from the paper).
>
> Mitigation strategy: The mitigation strategy is indeed unlikely to address the issue where the LLM does not know the answers (e.g., has not seen them during training). Our focus is on looking at the reasoning capabilities of LLMs via answer-set consistency, rather than the completeness or correctness of answers, which may be more dependent on training data. The issues of completeness / correctness are well-covered by existing benchmarks, and we see our work as complementing (certainly not replacing) these efforts to understand the capabilities of reasoning about such questions rather than responding to them. We clarify this in Section 3.3. There is an important caveat: empty results are trivially consistent, and thus we also extend the discussion on the “idk” (empty or i-don’t-know responses) in the results of Section 4.2 and in Table 3.
>
> Dataset construction: We were initially interested in testing equivalence and containment. We started with 600 pairs of questions $(Q_1,Q_2)$ that are equivalent (should return the same answers); for example, “What are the chambers of the human heart?” and “What cardiac chambers do people have?”. Next, when thinking of generating examples for containment, rather than starting from scratch with brand new questions, to reduce manual effort, we rather extended the pairs for equivalence, adding a $Q_3$, e.g., “What are the right-hand side chambers of the human heart?” whose answers were contained in $Q_1$. While creating these $Q_3$ questions, we realised that it would require relatively low effort to add the question $Q_4$ capturing the results of $Q_1$ not in $Q_3$, e.g., “What are the left-hand side chambers of the human heart?”, which would also allow us to test disjointness. This led to creating our datasets with quadruples of the form $(Q_1,Q_2,Q_3,Q_4)$. We have revised the description of the dataset construction in Section 3.2 to hopefully clarify the rationale behind our quadruples: in summary, they balance manual effort creating the dataset with the variety of relations we can test for answer set inconsistency.
>
> Missing examples: We have added Examples 3.1 and 3.2 to help to clarify these issues, and we have embedded further examples (in relation to Table 1) into our discussion. We hope that these improve readability (and would be happy to add more examples if needed).
>
> (Please note that we have uploaded a revised version of our submission for the reviewers’ consideration, with changes to address reviewer comments highlighted in blue.)

---

> > ### Comment · Reviewer_Cckv · 2025-11-26
> >
> > Thanks to the authors for their rebuttal and for making substantial changes to the paper.
> >
> > The notation for the datasets as a quadruple is still a bad choice, since if you later want to add a new question, the notation changes.
> >
> > Regardless of that, the main paper now looks substantially different from the originally submitted version. I am still keeping my original score, since I am not certain to what extent paper/results can be modified.

---

> > > ### Comment · Reviewer_Cckv · 2025-11-26
> > >
> > > There can be deeper analysis on the root cause of such inconsistencies, which the paper, in its current form, has neglected. If the LLM knows the answer and still is inconsistent, then it becomes a problem. But when the LLM does not know the answer, the current estimation of inconsistency is a joint factor of LLM's not knowing the answer + being inconsistent. Is it possible to disentangle the two?
> > >
> > > Such analysis can be insightful.
> > >
> > > One possible direction is to make sure that the required information to answer the question is present in the context. And then check if the LLM is consistent or inconsistent.

---

> ### Author Response · Authors · 2025-11-27
> **Reply to Cckv's concerns**
>
> We appreciate your additional comments, but we would like to clarify a few points, as there are might misunderstandings.
>  ## On the “quadruple notation” concern.
> We didn’t see why adding a new question needed to change the notation. The current dataset consists of 600 quadruples, and it is extendable to any number of them. When there are 5th, 6th or even more questions in each question set,
> it is still easy to extend and does not hurt the current notations. I can show you here:
>
> For a question set with $n$ questions, for any given question $Q_i$ and $Q_j$, the relation between $Q_i$ and $Q_j$ is $R \in$ { $E$, $C$, $D$, $O$ } (equivalence, containment, disjoint, overlap), and we can denote the relation as $R_{i,j}$.
>
> For example, $Q_5$ as an extended question that is equivalent to $Q_4$, we denote the relation between them as $E_{4,5}$.
>
> If you think that using the term “quadruple” is misleading to the scalability, we are willing to change it. If this is not what you concerned, could you please further explain it?
>
>  ## The revised paper is “substantially different”..
> I guess you may have this concern based on the appearance of change (indeed there are a lot of blue from longer explanations, clearer structure), not actual changes to the method or results. As the weaknesses include the notations, and ask rewriting to some paragraphs, adding examples, we thoroughly considered all of these suggestions, resulting in such 'a lot' difference.
> However:
>  - **All technical contributions from original submission are unchanged.**
>  - **All experiments and results consistent with the original submission.**
> - **Most changes are editorial**: clarification of notation, additional examples, reorganization for readability. We improved the notation in the revision precisely because reviewers asked for more clarity, but the core semantics did not change.
>
> For these results: as reviewer **KUx5** pointed out that some of the questions are not subjective, we carefully reviewed all the questions, and revised **10 questions** in total, and ran experiments on these 10. Thus, the overall analysis results are affected, but **this does not hurt the original conclusion**.
>
>
>
> ## causes of inconsistencies, and distinguish not knowing the answer + being inconsisteny.
> The causual analysis of inconsistencies are discussed in line 352-364, 396-407 and Appendix G, H.
> We also already distinguishes the two sources of inconsistency that you mentioned, see line 324-344. Specifically:
>
> - For questions that LLMs  “does not know” the answer is quantified through the %IDK rate, capturing cases where the model expresses uncertainty or inability to answer.
> - In both Consistency rates and Jaccard similarity measures, we excluded empty answer sets and responses of “idk”, which are reported separately.
> - Genuine inconsistency is captured through the $E(1,*)$ metric, which measures inconsistency by answering the same questions twice.
>
>
> **Comments**: Thanks a lot for your reply, but please don't be misled by the "a lot" blue text. Please read through the paper, you will find that its content is consistent with the original submission. We are looking forward to your response!

---

> > ### Comment · Reviewer_Cckv · 2025-11-27
> >
> > Thanks for the efforts on continuously improving the paper.
> >
> > Regarding the notation of dataset, there is an assumption that four kinds of manipulation are possible in the case of answering as a set: equivalence, containment, disjoint, overlap, and hence the consideration of a quadruple. Is there a principled understanding that these four are necessary and sufficient? My suggestion is to either show such completeness formally or keep the notation flexible. The tuple-centric notation considers a fixed construction, if I am not wrong.
> >
> > Causes of inconsistencies look like a subjective opinion, and without any solid experimental evidence. Line 352 to 358 mentions two reasons, while Appendix G has four reasons.
> >
> > Can you rephrase your original rebuttal on *mitigation strategy for inconstancy*? Are you claiming that this is not the focus of this work? A complete paper should raise a problem, and provide a solution. Currently, the focus has shifted heavily to saying that LLMs are not consistent. What about solutions?

---

> > > ### Author Response · Authors · 2025-11-27
> > > **Reply to Reviewer Cckv regarding the notation of dataset.**
> > >
> > > Thanks so much for your fast response!!!
> > >
> > > **Regarding notations of dataset**
> > >
> > > Our investigation focuses on consistency based on *answer sets*, where evaluation across four basic set relations—**equivalence**, **containment**, **disjointness**, **overlap**, and an advanced **minus**.
> > >
> > > When we examine the genuine inconsistency of LLMs, we consider cases caused by **stochasticity in the generation process** (lines 352–358), along with the four root causes summarized in Appendix G. In this setting, we only need the model to answer the same question repeatedly. We denote this as:
> > > $E_{1,*}$ (or equivalently $E_{1,1}$).
> > >
> > >
> > > We also analyze consistency across different formulations of semantically equivalent questions. For this purpose, we introduce $Q_2$.
> > >  $Q_1$ and $Q_2$ are semantically equivalent, so their answer sets should also be equivalent. We measure this using:
> > > $E_{1,2}$, reads as "if the the answer sets of $Q_1$ equivalent to that of $Q_2$".
> > >
> > >
> > > This is different from $E_{1,1}$: even though the semantics match, the token order changes, requiring the LLM to recognize semantic relaton between $Q_1$ and $Q_2$. The difference between $E_{1,1}$ and $E_{1,2}$ reflects the impact from (2) **semantic misunderstanding**, as discussed in lines 352–364. This attribution is not subjective—it is commonly exist for LLMs and even human beings.
> > >
> > > Beyond equivalence, we also examine containment. For this, we introduce $Q_3$, which is semantically subsumed by $Q_1$ (and $Q_2$). To verify whether this subsumption holds in the LLMs answers, we have:
> > > $N_{3,1}$ — reads as whether the answer set of $Q_3$ is narrower than that of $Q_1$.
> > >
> > >
> > > We further investigate more challenging relations, including disjointness and minus. We introduce $Q_4$, constructed so that the answer set of $Q_1$ equals the union of $Q_3$ and $Q_4$, and that $Q_3$ is disjoint from $Q_4$. This enables evaluation of:
> > > $D_{3,4}$ — reads as whether the answer sets of $Q_3$ and $Q_4$ are disjoint;
> > >
> > >
> > > $E_{3,1\setminus 4}$ — reads as whether the answer set of $Q_3$ equals the answer set of $Q_1$ minus $Q_4$.
> > >
> > >
> > > In this setup, we systematically evaluate equivalence, containment, disjointness, and minus. Each question in our quadruple is necessary for covering one of these relations. Here overlap is skiped as it always holds in equivalence and containment.
> > > While we could test additional relations such as $N_{4,2}$, they would not introduce new relational types beyond those already explored. Similarly, we could extend to $Q_5, \dots, Q_n$, but all relevant set relations are already included, we are not interested to go further in this direction. However, the design of notations remains extensible through the general notation:
> > > $R_{i,j}$
> > >
> > >
> > > Therefore, we consider our quadruple is necessary and sufficient for our evaluation. Related descriptions appear in Sections 3.1 and 3.2.

---

> > > > ### Author Response · Authors · 2025-11-27
> > > > **Reply to Reviewer Cckv regarding Causes of Inconsistency and Mitigation.**
> > > >
> > > > **Causes of Inconsistency**
> > > >
> > > > I disagree with the claim that causes of inconsistency are subjective and lack experimental evidence. In lines 352–358, we attribute inconsistency to:
> > > > 1. **Stochasticity**
> > > > 2. **Semantic misunderstanding**
> > > > I think this what we can do as far as we know, do you have any other idea?
> > > > Although you argued what if the LLMs don’t know the answer, we control for cases where the LLM does not know the answer by using **%IDK**, excluding such cases from inconsistency measures. Even though technically *%IDK** itself is consistent.
> > > >
> > > > Resolve the root cause of (1) stochasticity is not our main focus. It is related to “non-determinstic” of LLMs, and there are quite some related work with experiments, that are well-documented in related work and Appendix G, H. On top of that, we try to assess and mitigate the impact from (2), thus we designed **Classification-then-Enumeration (CtE)** and **Oracle strategies**. There are also further experiments on this are presented in Appendix F, called  **Answer-Set Contradictions**.
> > > >
> > > > **Mitigation**
> > > >
> > > > Indeed, our mitigation strategy operates at the conversational level, and we show that it already improves consistency while allowing us to attribute the underlying causes of inconsistency. We do not explore parametric strategies such as retraining or fine-tuning, although such methods may help in principle.
> > > > However, the most powerful LLMs evaluated in this work are closed models, so parametric interventions are not feasible. Our focus is therefore on analyzing the causes and applying non-parametric mitigation where possible.
> > > >
> > > > Regardless of it, as we mentioned before, this refer to **non-deterministic** of LLMs, where related works and appendix H also illustrated.
> > > > Eliminating inconsistency entirely would require making model behavior **deterministic**, but deterministic methods typically come with a trade-off: they often reduce answer quality, which is undesirable for generative models. It is important to acknowledge that modern LLMs are inherently non-deterministic, and although some recent work attempts to “eliminate” such variability, these approaches tend to degrade performance and undermine the benefits of generative modeling.
> > > > Again, we are not trying to eliminate the inconsistency, as the root cause come from the nature of Generative models. Make a generative model deterministic is not we are interested.
> > > >
> > > > Regarding the concern about “IDK,” we explicitly separate and measure it using the %IDK metric, ensuring that uncertainty is not conflated with inconsistency.

---

> ### Comment · Reviewer_Cckv · 2025-11-27
>
> I have updated the score. But I will not champion the paper for the following reasons.
>
> - I do not fully agree with the argument on inconsistency mitigations. I fail to see how determinism and inconsistency are related. A model can be deterministic yet inconsistent. The argument that powerful models cannot be pretrained is vague. If pretraining helps, it is better to showcase on some smaller models to make a strong point.
>
> - There should be a formal statement on the coverage of set relations. Union, intersection, and set-difference seem to be the primitive set relations. The paper already covers them. But it needs to be formalized. Otherwise, one would argue whether the setup is sufficient or not.
>
> - A less important point. When the LLM says "IDK", we are perhaps making an over-interpretation. If we really believe that the LLM says "I don't know" by understanding the question, they are already consistent to some extent. However, this is not the main point of the paper, and one paper does not need to answer everything.
>
> And, please refrain from answering very long replies. It is hard to get your point clearly.

---

> > ### Author Response · Authors · 2025-11-28
> > **Reply to Reviewer Cckv for "inconsistency mitigations"**
> >
> > We appreciate it very much that you are willing to raise the score and your fast response!
> >
> > Regarding your remaining concerns, I would like to response to them individually.
> >
> > I apologize that this maybe not short, we try to make it easier to read by adding main points for each paragraph.
> >
> > ## **Inconsistency mitigations**
> >
> > ***Stochasticity links non-determinism and inconsistency.***
> >
> > Since stochasticity is a common root cause of both inconsistency and non-determinism, we consider them related. We agree that a deterministic model can still be inconsistent, but our point is that non-determinism is a sufficient condition for stochastic inconsistency. If a model gives non-deterministic (and thus
> > different) answers to $Q_1$ at different times, it is also likely to
> > give different (i.e., inconsistent) answers for $Q_1$ and $Q_2$ as
> > equivalent queries.
> >
> > ***We evaluate the two main causes separately.***
> >
> > We divide inconsistency into two causes:
> >
> > (1) *stochasticity*, and
> >
> > (2) *semantic misunderstanding*.
> >
> > Non-determinism guarantees Cause (1). Enforcing determinism would remove Cause (1) and isolate Cause (2), but it would also change the scale of Cause (2) due to interactions with pre-training or fine-tuning. This would make the analysis of Cause (2) less accurate. Therefore, instead of forcing determinism, we use a subtraction-based approach.
> >
> > ***Cause (1) is measured via repetition; Cause (2) via excess inconsistency.***
> >
> > We measure the inconsistency from Cause (1) with $E_{1,1}$ by repeatedly asking the same question. We then estimate Cause (2) by comparing the inconsistency between questions (e.g., $E_{1,2}$, $N_{3,1}$) against $E_{1,1}$. The ``excess'' over $E_{1,1}$ is attributed to Cause (2). In the CtE and Oracle experiments, we further analyze Cause (2) by comparing these residuals with those of the Base.
> >
> > ***Fine-tuning can reduce inconsistency, but we skeptical wether its worthy.***
> >
> > We acknowledge that fine-tuning or other parametric strategies can reduce inconsistency, and related works have shown this. Therefore, we are skeptical if this something we should prove again in our work, altough this is not our focus.
> >
> > Fine-tuning often improves task-specific performance but reduces generality. It also requires high-quality datasets; as discussed in Appendix~G.4, better training data can mitigate inconsistency, but this leads in a different direction that is outside the scope of this work. Indeed, we could fine-tune smaller LLMs and expect better consistency, but we are skeptical about the practical gains.
> >
> > ***Our mitigation strategies are the better choice in general.***
> > We consider our mitigation strategies to be meaningful because:
> >
> > (1) they provide significant improvements in statistics;
> >
> > (2) they require neither large compute cost nor high-quality datasets; and
> >
> > (3) they generalize to all LLMs, including closed-source ones.
> >
> > If helpful, we can clarify these points in Appendices G and H, where we discuss the causes of inconsistency.
> >
> > **We are exciting to know how you think!**

---

> > > ### Author Response · Authors · 2025-11-28
> > > **Reply to Reviewer Cckv for "formal statement on the coverage of set relations"**
> > >
> > > We are still unsure what such a formal statement would look like.
> > > Whatever set relations one considers, one could always formulate a new
> > > set relation.
> > > We will keep thinking about it.
> > > However, if it is possible, it would help greatly to provide an example of a formal
> > > statement (or an idea in that direction) that would help us to prove the
> > > sufficiency of our setup.

---

> > ### Author Response · Authors · 2025-11-28
> > **Reply to Reviewer Cckv for "IDK"**
> >
> > Regarding "IDK", we agree that this can interfere with
> > (in)consistency. For this reason we present IDK rates separately.
> > We also discuss the trade-off between %IDK and consistency without IDK.
> > In short, different prompting strategies may encourage an LLM to answer IDK when it is uncertain. This increases consistency but at the cost of producing more IDK responses. Although IDK is always consistent by definition, it is usually not desirable in question-answering tasks. Hence, we treat this effect as a trade-off rather than a straightforward improvement.

---

> ### Author Response · Authors · 2025-11-29
> **Upate to Reviewer Cckv for "formal statement on the coverage of set relations"**
>
> Here we try to give a formal statement. If you think this is the right direction to go, please let us know!
>
> ## **Formal Coverage of Set Relations**
>
> Let $\mathcal{A}$ be the answer universe, the set of all possible atomic answers.
>
> Let $\mathcal{Q}$ be a set of emulation questions, $|\mathcal{Q}| = n$.
>
> Each question $Q \in \mathcal{Q}$ maps to a finite answer set, denoted as $||Q||$, where $||Q|| \subseteq \mathcal{A}$.
>
> For any pair of questions $(Q_i, Q_j)$, where $||Q_i|| \subseteq \mathcal{A}$ and $||Q_j|| \subseteq \mathcal{A}$, let $R_{i,j}$ denote the set relation between $||Q_i||$ and $||Q_j||$.
>
> We compactly denote the resulting relation as $R_{i,j} \in$ {$\equiv, \subset, \supset, \bowtie, \bot$}.
>
> The following five relations form a complete and mutually exclusive partitioning of all possible pairwise relations between two finite sets $||Q_i||$ and $||Q_j||$.
>
> **($\equiv$) Equivalence**, denoted as $E$:
>
> The answer sets are identical.$$||Q_i|| \equiv ||Q_j|| \iff ||Q_i|| = ||Q_j||$$
>
> **($\subset$ or $\supset$) Subsumption/Containment**, denoted as $C$:
>
> One answer set is a proper subset of the other.
>
> $$||Q_i|| \subset ||Q_j|| \iff ||Q_i|| \subsetneq ||Q_j||$$$$\text{or }$$$$||Q_i|| \supset ||Q_j|| \iff ||Q_j|| \subsetneq ||Q_i||$$
>
> **($\bowtie$) Overlap**, denoted as $O$:
>
> The answer sets share common elements but neither is a subset of the other.
>
> $$||Q_i|| \bowtie ||Q_j|| \iff \left(||Q_i|| \cap ||Q_j|| \neq \emptyset\right) \land \left(||Q_i|| \not\subseteq ||Q_j||\right) \land \left(||Q_j|| \not\subseteq ||Q_i||\right)$$
>
> **($\bot$) Disjointness**, denoted as $D$:
>
> The answer sets share no common elements.
>
> $$||Q_i|| \bot ||Q_j|| \iff ||Q_i|| \cap ||Q_j|| = \emptyset$$
>
> Use these types of relations are sufficient and necessary to capture all possible relationships between two finite answer sets. Our dataset construction ensures these relations are thoroughly covered, allowing us to evaluate LLM consistency across the spectrum of answer-set relations.
>
> To investigate LLM consistency across these relations, we construct each instance of our dataset as a quadruple
> $\{Q_1, Q_2, Q_3, Q_4\}$. Specifically, we use
> $R_{1,2} = E$, $R_{1,3} = C$, $R_{1, 4} = C$, $R_{3, 4} = D$, and one more advanced set minus $R_{4, 1\setminus3} = E$, ( $||Q_4|| \equiv (||Q_1|| \setminus ||Q_4||)$).

---

### Official Review · Reviewer_RNjR · 2025-10-31

**Soundness:** 3
**Presentation:** 2
**Contribution:** 2
**Rating:** 6
**Confidence:** 3

**Summary:**

In this paper, the authors introduce answer-set inconsistency, in which large language models (LLMs) can produce contradictory answers to enumeration questions that should obey set-theoretic relations.
In this paper, the authors have focused on the set relationship such as equality, containment, and disjointness between question-answer sets.
To evaluate it, they constructed the Answer-Set Consistency Benchmark (ASCB) containing 600 handcrafted quadruples (a total of 2,400 questions) derived from knowledge-graph QA datasets.
Three approaches are defined 1) Base, 2) Classify-then-Enumerate (CtE), and 3) Oracle and these approaches are used to assess both relation classification and consistency of generated answers.
Experiments on 18 modern LLMs, including GPT-5, Gemini-2.5, and Llama-3, reveal pervasive inconsistency even under low temperature (greedy sampling) settings.
Containment and multi-set relations are found to be the most difficult, while equivalence and disjointness are easier.
The CtE strategy, which prompts the model to reason about relations before answering, significantly improves consistency across all models.

**Strengths:**

S1. The paper clearly formalizes an important concept, LLM answer-set consistency. This concept is quite interesting and has potential to provide a principled framework for LLM to be more logical coherent.

S2. The empirical analysis across 18 modern models and the 2400 questions dataset are valuable contributions to understand LLM behaviours.

**Weaknesses:**

W1. The proposed mitigation strategies, such as Classify-then-Enumerate, are relatively straightforward and simple. It would be great if the authors can provide more fundamental algorithmic improvements or mitigation strategy.

W2. While the paper identifies sources of inconsistency, it provides limited theoretical analysis or deeper causal explanation beyond empirical observation.

W3. The symbols in this paper feels a bit overloaded. $Q_i$, $RQ_i$, $R_i$. Maybe create a symbol table for line 280-285 such that the readers can find the meaning of the different relationship easier.

Comment: The author may find the following papers related to set/list semantic equivalence to be interesting: https://arxiv.org/abs/2312.10321 and https://arxiv.org/abs/2502.12466.

**Questions:**

Please see W1-W3.

---

> ### Author Response · Authors · 2025-11-21
> **Reply to Reviewer RNjR for W1-W3**
>
> We appreciate your valuable comments and time, and the opportunity to improve our work. We respond to the three weaknesses (W*) and the comment (C) indicated:
>
> W1: Though indeed simple, the mitigation strategies lead to statistically significant improvements, which we view as a positive result. Better improvements could be achieved through more complex strategies, which we agree is an interesting direction for future work. One such idea, which we now present in future work, would be to parse the question into a structured representation, and then using the LLM (or perhaps even an external service) to decide the relations between these representations. An important caveat here is that this can become undecidable for sufficiently complex questions. Some other strategies that may be able to mitigate the inconsistency of LLMs are discussed in Appendix G and H.
>
> W2: We are working on this, and delve into more details to understand potential reasons for inconsistency in the new Appendices G and H. We have also added some additional discussion to summarize these points better in Section 4.2 in the body of the paper. Given the stochastic nature of LLMs, a more theoretical analysis is challenging, except to say that the underlying tasks of query containment, equivalence, etc., are known to be undecidable for structured query languages like SQL.
>
> W3: We have significantly revised the paper to provide a simpler notation, and have added Table 3 with an explanation of the notation we introduce. We hope this improves clarity and readability.
>
> C: We appreciate the pointers to these papers, which are indeed very related, and have been added to our discussion of related works. Indeed, as mentioned in our future work, LLM-SQL-Solver could be a useful basis (relating to W1) for a mitigation strategy that parses and reasons about structured representations of questions.
>
> (Please note that we have uploaded a revised version of our submission for the reviewers’ consideration, with changes to address reviewer comments highlighted in blue.)
>
> **Thank you again for your valuable feedback! Since the rebuttal window closes in just a few days, it would be great if we could have a conversational discussion. We truly appreciate your time and look forward to your response!**

---

### Official Review · Reviewer_KUx5 · 2025-11-05

**Soundness:** 3
**Presentation:** 3
**Contribution:** 2
**Rating:** 4
**Confidence:** 5

**Summary:**

This paper proposes to study a novel concept of answer-set consistency in large language models (LLMs), which refers to whether an LLM's responses to related enumeration questions respect expected set-theoretic relations such as equality, containment, and disjointness. The authors introduced a new benchmark with 600 handcrafted quadruples of logically related questions and used it to evaluate 18 contemporary LLMs under several prompting strategies. As evaluating, explaining, and addressing answer-set inconsistency in LLMs' question-answering capability has immense practical value, the paper is quite valuable in this regard.

**Strengths:**

1. **Novelty:** The paper makes a novel contribution by formalizing the concept of answer-set inconsistency for enumeration questions. This establishes a new, well-defined dimension for evaluating LLM's reliability based on set-theoretic relations.

2. **Thorogh empirical analysis**: I appreciate the paper's comprehensive and systematic empirical analysis, which helps understand the nature of answer-set inconsistencies across a wide variety of LLMs.

**Weaknesses:**

1. **Objectivity of some questions in the benchmark**: Some of the questions in the dataset are not objective.
For instance, “On what video streaming services can I watch the Hunter x Hunter anime series?”-it depends on the region or country.

2. **Incompatiblity of Problem formulation and benchmark**: In section 2.1, the authors assume E to be "the universe of entities from an arbitrary domain" from where both questions and answers originate.  This formulation implicitly assumes an open-world or extensible entity space -- that is, E could, in principle, include all possible entities that exist within a domain, even those not observed in the dataset. Under this assumption, answer-set consistency is defined in an idealized, domain-agnostic logical sense. But, in practice, the ASCB benchmark is constructed from knowledge graphs (KGs) such as Wikidata and DBpedia, which operate under a closed-world assumption. Meaning,
the benchmark enumerations are derived only from entities present in the KG; Missing facts or unobserved entities are treated as false rather than unknown; Consequently, the "universe"  E is finite and closed to the KG’s vocabulary. This means that while the paper’s definitions rely on the notion of “true set relations within an arbitrary domain,” the evaluations measure consistency only relative to KG-encoded truth -- not the broader, theoretically open universe.

3. **Lacking discussion on Synthetic**: Please discuss in detail the construction process of SYNTHETIC, in particular, what the prompts were, and how you came up with the questions in the first place.

4. **Deeper understanding of the results is missing:** Table 3, ~R4 column => Why CtE and Oracle are underperforming (lower score) compared to Base in many models? You identified this issue in Appendix E, but are there some inherent limitations to your mitigation strategies on disjoint set queries?  We also see this problem on GPT5 ~R5, CtE. In addition, we also see that on %R5, CtE strategy makes GPT-4.1-nano, and GPT-5-nano both perform worse. What are the reasons behind this?

**Questions:**

See W2 and W4.

---

> ### Author Response · Authors · 2025-11-21
> **Reply to Reviewer KUx5 for W1-W4**
>
> We are grateful for your valuable comments and time, and the opportunity to improve our work. We respond to the four weaknesses indicated:
>
> W1: We agree that the answers to the question highlighted may differ depending on the region. We have revised all of the questions for subjectivity and contextual sensitivity and identified 10 questions (e.g., another being “Which television shows are shot in New York?”) that we have revised. We have updated the experimental results on all models with these revised questions and have updated the paper with them. They have led to some minor changes to results, but no changes to our main findings.
>
> W2: We agree that this was unclear. Our intention with $\textbf{E}$ was to define the set of all possible answers (strings) to bootstrap the set-based definitions. Our analysis is agnostic to the entities or domain in this latter sense, where we look only at the internal set consistency between the answer sets for different questions for a given model. More specifically, for a given model, if we were to map every answer (over all questions) to an arbitrary fresh answer in a one-to-one manner, our results would remain unchanged since this preserves answer set consistency (relations like set equivalence, containment, disjointness, etc., are preserved). With one caveat, our results are also agnostic to CWA/OWA, as all relations are defined with respect to the results given by the model for $Q_1$: $Q_2$ is equivalent to $Q_1$ and $Q_3$ and $Q_4$ form a strict dichotomy of the results of $Q_1$. The caveat here is the case when $Q_1$ is empty due to incompleteness (OWA), which we analyze separately. We have simplified the definitions in Section 3.1, clarified why we use KGQA datasets in Section 3.2, and discussed more explicitly the empty/idk cases in Section 4.2.
>
> W3: The questions themselves were generated by the LLM, along with their variants. We appreciate the need to be clearer on this, and provide more details in Appendix B of the revised submission.
>
> W4: We clarify that $\sim$ referred to Jaccard similarity, and that $R_4$ referred to disjointness. Thus a lower Jaccard similarity is better for disjointness (where 0 indicates disjoint). Furthermore, we recall that CtE asks the model to classify the relation between two questions before enumerating their answers; we expect that this will only improve results if the model can classify the relation. The classification results for GPT-4.1-nano for $R_5$ (now $E_{4,1\setminus3}$), as seen in Table 5, are poor (it always returns ‘idk’), which explains why the result for CtE is also worse. Another issue is that models sometimes get confused by the more complex prompts: for %$R_5$ (now $E_{4,1\setminus3}^\text{con}$)  of GPT-5-nano, it often returns relations (as requested in the first step) instead of returning answers to the questions in the second step. For ${\sim}R_5$ (now $E_{4,1\setminus3}^\text{sim}$) of GPT-5, we found a typo for Base, which should be 0.4804, rather than 0.6792 (we appreciate a lot for highlighting this case), which is now updated.
>
> (Please note that we have uploaded a revised version of our submission for the reviewers’ consideration, with changes to address reviewer comments highlighted in blue.)
>
> **Thank you again for your valuable feedback! Since the rebuttal window closes in just a few days, it would be great if we could have a conversational discussion. We truly appreciate your time and look forward to your response!**

---

> ### Comment · Reviewer_KUx5 · 2025-11-27
>
> Thank you for your response.
>
> Re-W1. How many quadruples (among the 600) were affected by this change? Could you explain in detail the exact process you used to filter questions for "subjectivity and contextual sensitivity"? Manually/with LLM? If manually, how many human evaluators validated/vetted the outcome?
>
> Re-W2. The write-up is much clearer now. There are still some questions in this discourse about OWA - CWA and how that relates to finite and countable finite answer sets. To my understanding, the assumption is that [[Q]] is finite, and you do not allow questions whose answer-set is countably infinite, for instance, Q1: What are the primes > 100? Could you confirm this to be correct/incorrect?
>
> Re-W3. Upon re-reading Section B.2 (QUESTION PIPELINE) and given that you had some subjective questions that needed manual post-processing/sanity checks during rebuttal, I am skeptical of this agentic pipeline. Have you manually validated/vetted the generation of Q3 and Q4? Have you manually validated/vetted the semantic relation between all 4 questions in each quadruple? If not, relying on Agent 4 undermines the credibility of the quadruples (in particular, Q3 and Q4) in your benchmark.
>
> **W5 (In reference to Reviewer Cckv's point on mitigation strategy, Reviewer RNjR also pointed to this issue).** I do agree that a better mitigation strategy is needed. In particular, we do see similar works that addressed inconsistencies in LLMs showed that parametric mitigation strategies (e.g., finetuning) helped (See Ghosh et al., ICLR'25, in your reference). Therefore, I would also encourage the authors to go beyond the conversational level and showcase a more robust mitigation strategy for some open-source models to make a stronger point and enhance the paper's contributions.

---

> > ### Author Response · Authors · 2025-11-28
> > **Reply to Reviewer KUx5 for updated concerns.**
> >
> > Thanks a lot for the additional feedback following our response!
> >
> > Now I response to them accordingly.
> >
> > **Re-W1**: A total of 10 questions were modified. Because each question belongs to a predefined set of four semantically related questions, revising one question required updating the entire corresponding set, resulting in forty revised questions overall. Given the time constraints, this refinement was conducted manually by one of the authors. The filtering procedure addressed issues of subjectivity and contextual sensitivity. Specifically, we removed or rephrased questions that depended on geographical context (such as the example you initially proposed) or that could become ambiguous without an explicit temporal reference. For instance, the question “Give me all soccer clubs in the Premier League” was modified to “Give me all soccer clubs in the Premier League for the 2023–2024 season.” Although the current revision was performed by a single author, we would be pleased to develop a more robust methodology involving four evaluators, three authors and a state-of-the-art LLM, to further refine the dataset. Nonetheless, the applied criteria align with the principles of the original dataset curation, and based on our follow-up analysis, we expect only negligible effects on the results and no impact on the validity of our claims or their statistical significance.
> >
> > **Re-W2**: We appreciate that the reviewer find the write-up cleaner. Indeed, we upper-bound the number of results for this benchmark to 100 to avoid context-window effects, so asking to enumerate primes greater than 100 would be ruled out. Such questions might be addressed by a different methodology. One idea might be to sample answers and ask if they comply with different questions, e.g., if "is 2027 a super-prime?" should be true if and only if "is 2027 a prime?" is true. This could be a good way to address questions with countably infinite answers in future.
> >
> > **Re-W3**: The output generated by the question pipeline was evaluated by three authors of this article, and it was verified that the relationships between the sets of answers to the questions were respected. The question pipeline was mainly used to eliminate questions from the two starting datasets  LC-QUAD 2.0 and QALD) that were obviously not relevant to our purposes (e.g., where the answer was not a set of answers). The questions generated were modified, and often the output of the LLM was only a guide, but the questions were heavily modified or completely changed in some cases to meet the inclusion criteria. As a final step we also used LLMs to identify and correct possible issues in the manual refinement. We can further specify in the main text that the output of the question pipeline was not used as is, but went through the same revision process as the synthetic dataset. We believe the resulting questions and dataset are of high quality, and we make them available online for review: https://anonymous.4open.science/r/ASCS-7412/data/Dataset/en/
> >
> > **Re-W5**:  We agree with both reviewers that additional mitigation strategies could be explored, but we are skeptical about their necessity in our context. We have no doubt that strategies such as fine-tuning can reduce inconsistency; several prior works have already demonstrated this experimentally. This raises the question of whether have such experiments in our paper is necessary.
> >
> > Moreover, parametric-level strategies come with clear drawbacks:
> >
> > (1) they often improve task-specific performance at the cost of reduced generalization;
> >
> > (2) they require substantial compute resources and high-quality training data, while still offering limited generality; and
> >
> > (3) they cannot be applied to closed-source LLMs, which are currently the most capable models.
> >
> > In contrast, our mitigation strategies avoid these shortcomings and still yield consistent, statistically significant improvements. Additional points related to this discussion are addressed in our response to Reviewer Cckv under “Inconsistency Mitigations.”

---

### Author Response · Authors · 2025-12-02
**Summary Comment for the AC**

We summarize the actions taken to address all reviewer concerns.

## Reviewer KUx5

**[W1]** One author manually reviewed all 2,400 benchmark questions and revised 10 to resolve concerns about their objectivity. We re-ran the full experiment and updated the article; the adjustments produced minor numerical changes but did not affect the main findings. In follow-up discussion, we also specified the types of questions that were modified and provided illustrative examples.

**[W2]** We clarified that our results are agnostic to both the Closed-World and Open-World Assumptions. Definitions in Section 3.1 were simplified, the motivation for using KGQA datasets in Section 3.2 was made clearer, and the discussion of empty/“I don’t know’’ answers in Section 4.2 was expanded. We also directly addressed the example raised by the reviewer to further clarify the distinction between the two assumptions.

**[W3]** We added further details in Appendix B describing the construction of the synthetic dataset used in our answer-set consistency evaluation. Additionally, we expanded the explanation of our QUESTION PIPELINE, including the manual review applied to all questions filtered out by the pipeline. These filtered questions were independently examined by three authors.

**[W4]** We clarified the correct interpretation of the relevant table and corrected a typographical error identified by the reviewer.

## Reviewer RNjR

**[W1]** We emphasized that, despite their simplicity, our mitigation strategies deliver statistically significant improvements. We added an alternative mitigation strategy in the future work section and additional approaches for mitigating inconsistencies in Appendices G and H.

**[W2]** We expanded discussion of potential causes of inconsistency in Appendices G and H, and integrated a concise summary of these insights into Sec. 4.2.

**[W3]** To resolve multiple comments on notation, we simplified the notation across the paper and added Table 2 explaining all notation clearly.

**[C]** We incorporated the two referenced articles into the related work.

We have not yet received feedback.

## Reviewer Cckv

Following several rounds of constructive interaction, which led to a score increase from 2 to 4, the discussion is ongoing.

**[W1]** A major rewrite of Section 3 introducing simpler and more standard notation, together with Table 2 and additional examples. Other sections were revised for consistency.

**[W2]** Clarification that all prompts used to query LLMs were already included in Appendices A–B. We added details on LLM output parsing and on the evaluation pipeline.

**[W3]** Clarification that the focus of our work is assessing reasoning via answer-set consistency, not completeness or correctness, which are addressed by existing benchmarks. We expanded treatment of empty/IDK responses in Sec. 4.2 and Table 3.

**[W4]** Improved the dataset construction description (Sec. 3.2, Appendix B) with clearer rationale and additional examples.

**[W5]** Added Examples 3.1 and 3.2 and provided further illustrative examples in connection with Table 1.

Remaining concerns and our clarifications:

**[W1, W4]** Notation (“quadruples”): Reviewer argued the notation is unstable if new questions are added. We clarified with examples that the notation remains unchanged and is both necessary and sufficient for evaluation while flexible for extension. We also requested input on alternative notations and provided a revised proposal to gauge their expectations.

**[W3]** Root-cause analysis: We pointed out that deeper causal explanations now exist in the main text (lines 352–364, 396–407) and Appendices F–H. These analyses are experiment-supported and focused on semantic misunderstanding rather than stochasticity. Given prior work on pretraining/fine-tuning constraints and scope relevance, we do not consider additional experiments with such mitigation strategies necessary.

After these exchanges, concerns [W2] and [W5] are fully resolved. Concerns [W1] and [W4] are partially resolved, and [W3] was under discussion.

## Reviewer ykMU

**[W1]** We clarified that although the dataset contains only 2,400 questions, it is sufficient for statistically meaningful evaluation of static general-knowledge queries. We positioned scaling to dynamic settings as natural future work.

**[W2]** While our work focuses on single-turn interactions, our methodology readily supports multi-turn extensions. We added this as future work, as suggested.

**[W3]** We performed deeper error analysis and added extensive discussion in Appendices G and H, with expanded commentary in Sec. 4.2.

**[W4]** We clarified the dataset design objectives, construction rationale (Sec. 3.2, Appdenix B), and explained the choice of the five relations studied (Sec. 3.3).

The reviewer confirmed W3 is partially resolved. W1 and W2 remain, but we include them as future research directions enabled by our contribution. There is no comment for W4.

---

### Meta-Review · Area_Chair_Pri6 · 2026-01-06

**Summary:**

This work measures answer set consistency of LLMs responses over a set of related enumeration questions.
The reviewers recognize the novelty of the problem of set-theoretic relations in LLMs’ responses and the breadth of the evaluation, but have raised concerns about the value of the contribution given limited mitigation strategies proposals.

**Reviewer Concerns:**

See below.

**Reviewer Scores:**

Reviewer KUx5 raised concerns around the quality / ambiguity of questions in the dataset, details on synthetic data generation and mitigation strategies; the authors addressed most of their concerns (especially the clarifying questions) and I believe the reviewer would likely retain their score or increase to 6 (less likely), since the reviewer also agreed with concerns raised by other reviewers of lack of mitigation proposal.

Reviewer RNjR raised concerns regarding limited mitigation strategies beyond prompting based approaches and lack of theoretical analysis or explanations. I believe the rebuttal partially addressed the reviewer’s questions and they would likely retain score 6.

Reviewer ykMU raised concerns around 1) diversity of questions, 2) lack of multi-turn evaluation, 3) lack of error analysis. I believe that 1) and 2) are out of scope for this paper as the author’s rebuttal indicated. The authors didn’t fully address 3). The reviewer indicated they were going to retain their score 4 since they believed the questions were only partially addressed.

Reviewer Cckv raised the following concerns: 1) clarity of mathematical notation and dataset construction, 2) details on prompt construction, 3) mitigation strategy fixing the problem only on the surface level. The authors’ rebuttal partially addressed these concerns and the reviewer was going to raise the score 2->4 or 2->6 (less likely), but they indicated they didn’t want to champion the paper due to outstanding issues.

I believe that lack of theoretical justification shouldn't be a basis for rejection for a benchmarking paper. However, multiple reviewers also raised other concerns regarding the clarity of the paper, lack of error analysis, and the depth of the technical contribution on mitigation approaches. I believe the paper is borderline in the current form, but it will greatly benefit from another revision to address these outstanding concerns, thus I recommend a rejection.

---

### Decision · Program_Chairs · 2026-01-26

Reject